# Ab initio prediction of specific phospholipid complexes and membrane association of HIV-1 MPER antibodies by multi-scale simulations

Colleen A Maillie, Kiana Golden, Ian A Wilson, Andrew B Ward, Marco Mravic*

Department of Integrative Structural and Computational Biology, The Scripps Research Institute, La Jolla, United States

## eLife Assessment

This **valuable** study reports multi-scale molecular dynamics simulations to investigate a class of highly potent antibodies that simultaneously engage with the HIV-1 Envelope trimer and the viral membrane. The work provides insights into how broadly neutralizing antibodies associate with lipids proximal to membrane-associated epitopes to drive neutralization. After extensive revision, the level of evidence is considered **solid**, although a quantitative assessment of the underlying energetics remain difficult to obtain.

**\*For correspondence:**
mmravic@scripps.edu

**Competing interest:** The authors declare that no competing interests exist.

## Abstract

A potent class of HIV-1 broadly neutralizing antibodies (bnAbs) targets the envelope glycoprotein's membrane proximal exposed region (MPER) through a proposed mechanism where hypervariable loops embed into lipid bilayers and engage headgroup moieties alongside the epitope. We address the feasibility and determinant molecular features of this mechanism using multi-scale modeling. All-atom simulations of 4E10, PGZL1, 10E8, and LN01 docked onto HIV-like membranes consistently form phospholipid complexes at key complementarity-determining region loop sites, solidifying that stable and specific lipid interactions anchor bnAbs to membrane surfaces. Ancillary protein-lipid contacts reveal surprising contributions from antibody framework regions. Coarse-grained simulations effectively capture antibodies embedding into membranes. Simulations estimating protein-membrane interaction strength for PGZL1 variants along an inferred maturation pathway show bilayer affinity is evolved and correlates with neutralization potency. The modeling demonstrated here uncovers insights into lipid participation in antibodies' recognition of membrane proteins and highlights antibody features to prioritize in vaccine design.

## Introduction

Antibodies can target integral membrane proteins very near to the lipid bilayer surface, including at epitopes partially embedded within the headgroup region. During antigen engagement, antibodies complementarity-determining regions (CDR) may need to navigate lipid molecules to gain access to a concealed or dynamically buried site. Evolving tolerance or even affinity to lipid bilayers could be beneficial in recognition of membrane-proximal epitopes, potentially forming simultaneous cooperative interactions with antigen and membrane yielding additional avidity and specificity. Conversely, propensity to bind lipids or cell membranes poses a significant auto-immunity risk. B-cells producing antibodies targeting host membranes are downregulated in healthy organisms (*Goodnow et al., 2005*; *Liu et al., 2015*; *Verkoczy et al., 2010*; *Zhang et al., 2016*). Nonetheless, rare antibodies

with lipid affinity can emerge, particularly in cases of chronic inflammation and infection as in HIV (*Matyas et al., 2009*; *Alam et al., 2007*; *Alam et al., 2009*; *Haynes et al., 2005*). Maturation pathways of these rare events remain unclear, but must strike a careful balance of polyreactivity to avoid or overcome autoreactivity (*Caillat et al., 2020*). A better understanding of how antibodies develop membrane affinity and target membrane-proximal epitopes would be impactful for antibody therapeutics, auto-immunity, and vaccine development (*Burton and Hangartner, 2016*; *Mascola and Haynes, 2013*; *Krebs et al., 2019*; *Rujas et al., 2024*; *López et al., 2024*).

We sought to address this phenomenon for broadly neutralizing antibodies (bnAbs) 4E10, PGZL1, 10E8, and LN01 which stem from unique lineages and all target the semi-concealed membrane-proximal epitope region (MPER) (*Lee et al., 2016*; *Cardoso et al., 2007*; *Cardoso et al., 2005*; *Sun et al., 2008*) of the HIV-1 envelope glycoprotein (Env). Interestingly, these bnAbs show varying degrees of affinity for lipid components; some associate with lipid bilayers or cultured cells even in the absence of antigen (*Matyas et al., 2009*; *Alam et al., 2007*; *Alam et al., 2009*; *Haynes et al., 2005*, *Stiegler et al., 2001*; *Huang et al., 2012*; *Pinto et al., 2019*; *Zhang et al., 2019*; *Scherer et al., 2010*; *Chen et al., 2014*; *Yang et al., 2013*). This membrane interaction behavior appears to correlate with neutralization potency and is attributed to shared CDR loop features including a long hydrophobic CDR-H3 (*Alam et al., 2009*; *Huang et al., 2012*; *Scherer et al., 2010*; *Chen et al., 2014*; *Irimia et al., 2017*; *Kwon et al., 2018a*; *Zwick et al., 2004*). The correlation between membrane association and neutralization has been established, given that mutations or chemical conjugations to solvent-exposed antibody loop residues which enhance association to phospholipid vesicles consequently boost efficacy of pseudovirus neutralization (*Rujas et al., 2018*). Conversely, mutations reducing hydrophobicity drastically reduce neutralization activity while also weakening association to lipid bilayers, despite minimal impacts to antigen affinity (e.g. 4E10 CDR-H3 H100-H102 Trp-Trp motif; PGZL1 clone H4K3's CDR-H1; *Alam et al., 2009*; *Scherer et al., 2010*; *Xu et al., 2010*; *Julien et al., 2010*). Thus, phospholipid membrane interactions evolved by MPER-targeting bnAbs appear critical for their mechanism of immune protection against HIV. Yet, the breadth of molecular features responsible for association with membranes and mode of lipid-embedded epitope engagement are not clear.

Structural characterization of MPER-targeting bnAbs with full-length Env trimer or gp41 fragments also indicate that the surrounding lipid bilayer plays a role in antibody access and epitope recognition (*Lee et al., 2016*; *Rantalainen et al., 2020*; *Yang et al., 2022*). Cryo-electron microscopy (cryo-EM) of Env trimers bound to bnAbs PGZL1, 4E10, and 10E8 within in different model membranes suggest their CDR loops form intimate contacts with surrounding lipids while engaging MPER (*Rantalainen et al., 2020*; *Yang et al., 2022*). Antigen binding fragments (Fab) crystal structures of those bnAbs as well as LN01 soaked with short-chain phospholipids all revealed ordered headgroup moieties complexed within the CDR loops in the presence and absence of antigen (*Matyas et al., 2009*; *Haynes et al., 2005*; *Zhang et al., 2019*; *Irimia et al., 2016*; *Irimia et al., 2016*) suggesting the antibodies encode specific lipid interactions (*Matyas et al., 2009*; *Haynes et al., 2005*; *Zhang et al., 2019*; *Irimia et al., 2016*; *Irimia et al., 2016*). Thus, the molecular features mediating membrane affinity for these bnAbs appear critical to their maturation and mechanism of immune protection against HIV in vivo.

These data support a two-step bnAb neutralization mechanism proposed previously, wherein a population of bnAbs in vivo may first associate with membranes via embedding their CDRs, then laterally diffuse across the bilayer surface to subsequently engage Env at MPER (*Alam et al., 2007*; *Pinto et al., 2019*; *Chen et al., 2014*; *Irimia et al., 2017*; *Kwon et al., 2018a*; *Rantalainen et al., 2020*; *Irimia et al., 2016*; *Carravilla et al., 2020a*; *Soto et al., 2016*). The reverse order, MPER binding followed by CDR membrane insertion, is also often posited. Here, we developed multi-scale molecular dynamics (MD) simulation approaches suited to investigate these mechanisms and the unique maturation landscape these rare antibodies must navigate to avoid auto-immune consequences. For these MPER bnAbs, we focused on the in silico capacity to characterize the molecular features mediating phospholipid affinity at atomic detail, not afforded by previous structural approaches. We find that unbiased all-atom MD simulations accurately and reproducibly predict ab initio binding of phospholipids stably bound at specific CDR binding sites previously identified in co-crystal structures (*Pinto et al., 2019*; *Zhang et al., 2019*; *Irimia et al., 2017*; *Irimia et al., 2016*). Integrating coarse-grain (CG) simulations, we capture the full process of antibodies scanning and embedding into lipid bilayers to biologically relevant conformations. The simulations illuminate key molecular features

tuning bnAbs' lipid interaction and preferred membrane-bound geometry, advancing on previous work (*Carravilla et al., 2020a*), including surprising contributions from framework regions. Biased simulations provide rough estimates of membrane interaction strength, applied to demonstrate that in silico bilayer affinity correlates with neutralization efficacy of experimentally characterized PGZL1 variants along a pseudo-maturation trajectory (*Sun et al., 2008*).

This modeling platform provides an improved framework for navigating the proposed two-step bnAb neutralization mechanism both for retrospective and proactive evaluation of antibody repertoires, which should have utility in vaccine design and therapeutic antibody engineering. Likewise, our results inform more broadly generalized molecular principles for the in vivo selection of antibodies targeting membrane proteins, beyond HIV, at partially concealed juxtamembrane regions – desirable epitope regions given they often have high protein sequence conservation relative to water-soluble exposed sites. Reliable atomic detail views of antibody conformations at the membrane surface and loop phospholipid binding sites should be powerful for better understanding membrane protein epitope engagement, pathogen neutralization mechanism, and checkpoints regulating self-sensing antibodies.

## Results

### Atomic simulations accurately model MPER bnAb 4E10 and PGZL1 phospholipid interactions at HIV-like membrane surfaces

To assess the stability, organization, and spectrum of phospholipid interactions inferred from previous lipid-bound crystallography and cryo-EM experiments, we first performed unbiased all-atom simulations of 4E10, PGZL1, and 10E8 peripherally embedded to the surface of model lipid bilayers. Initial membrane-bound Fab conformations were computed using PPM2.0[3735] (*Lomize et al., 2012*), globally optimizing protein insertion based on per-residue hydrophobicity and solvation, each resulting in reasonable predictions with CDR-H3 embedded. Four 1 μs pseudo-replicate simulations were initiated per bnAb, with two replicates tilted by ± 15 degrees to modestly vary the starting membrane-interacting pose, utilizing an explicit simplified HIV-like model anionic bilayer: 25% cholesterol, 5% 1-p almitoyl-2-oleoyl-sn-glycero-3-phosphate (POPA), 70% 1-palmitoyl-2-oleoyl-sn-glycero-3-phosphocholine (POPC; *Alam et al., 2007*).

4E10 and PGZL1 are homologous (85% sequence identity, sharing IgG *VH1-69, VH4-34,* and *VH4-59*), and have very similar affinities, breadths, and potencies (paricularly comparing 4E10 and PGZL1. H4K3). 4E10 is notably more poly-reactive: binding to cultured cells, numerous isolated phospholipids (*Matyas et al., 2009*), and vesicles of most compositions even in the absence of antigen – whereas 10E8, LN01, and PGZL1 do not. PGZL1 and 4E10 crystal structures both bear putative lipid electron density at several surface loops, notably recurring within a CDR-H1 site – a region when mutation reduced the neutralization potency 10-fold for PGZL1-H4K3 (*Zhang et al., 2019*; *Irimia et al., 2016*; *Zwick et al., 2001*). Given their similarities, we first compared the lipid bilayer interaction mechanisms in 4E10 and PGZL1 simulations. Upon time-averaging phospholipid density from four independent trajectories, both bnAbs showed strong hotspots for a lipid phosphate bound within the CDR-H1 loops, with minimal phospholipid or cholesterol ordering around the proteins elsewhere. The simulated lipid phosphates bound within CDR-H1 have exceptional overlap with electron densities and atomic details of modelled headgroups from respective lipid-soaked co-crystal structures of 4E10 and PGZL1 (*Zhang et al., 2019*; *Irimia et al., 2017*; *Irimia et al., 2016*), with phosphate mean positions RMSDs of 0.6 and 0.7 Å, respectively, relative to the X-ray structures (*Figure 1A, B*; *Figure 1—figure supplement 1A–D*). Thus, our simulations reproduce ab initio with atomic accuracy the evidence of 4E10 and PGZL1 bnAbs' mechanism of membrane association. These results validate the CDR-H1 antibody-lipid interactions proposed from previous co-crystal structures are feasible in the context of full biologically realistic HIV-like bilayers and form very readily. The CDR-H1 lipid phosphate complexes appear to be stable, specific interactions that anchor and orient CDR loops at the bilayer surface for antigen engagement, complementing protein-lipid interactions which are less geometrically specific, that is lipid tail solvation of apolar groups in the CDR-H loops such as CDR-H3 tryptophans.

We next characterized the stability and dynamics of the spontaneously forming antibody-phospholipid complexes which recurred across all trajectories. For both 4E10 and PGZL1, three of four replicates showed very rapid binding of a headgroup phosphate ab initio in the first 20 nanoseconds

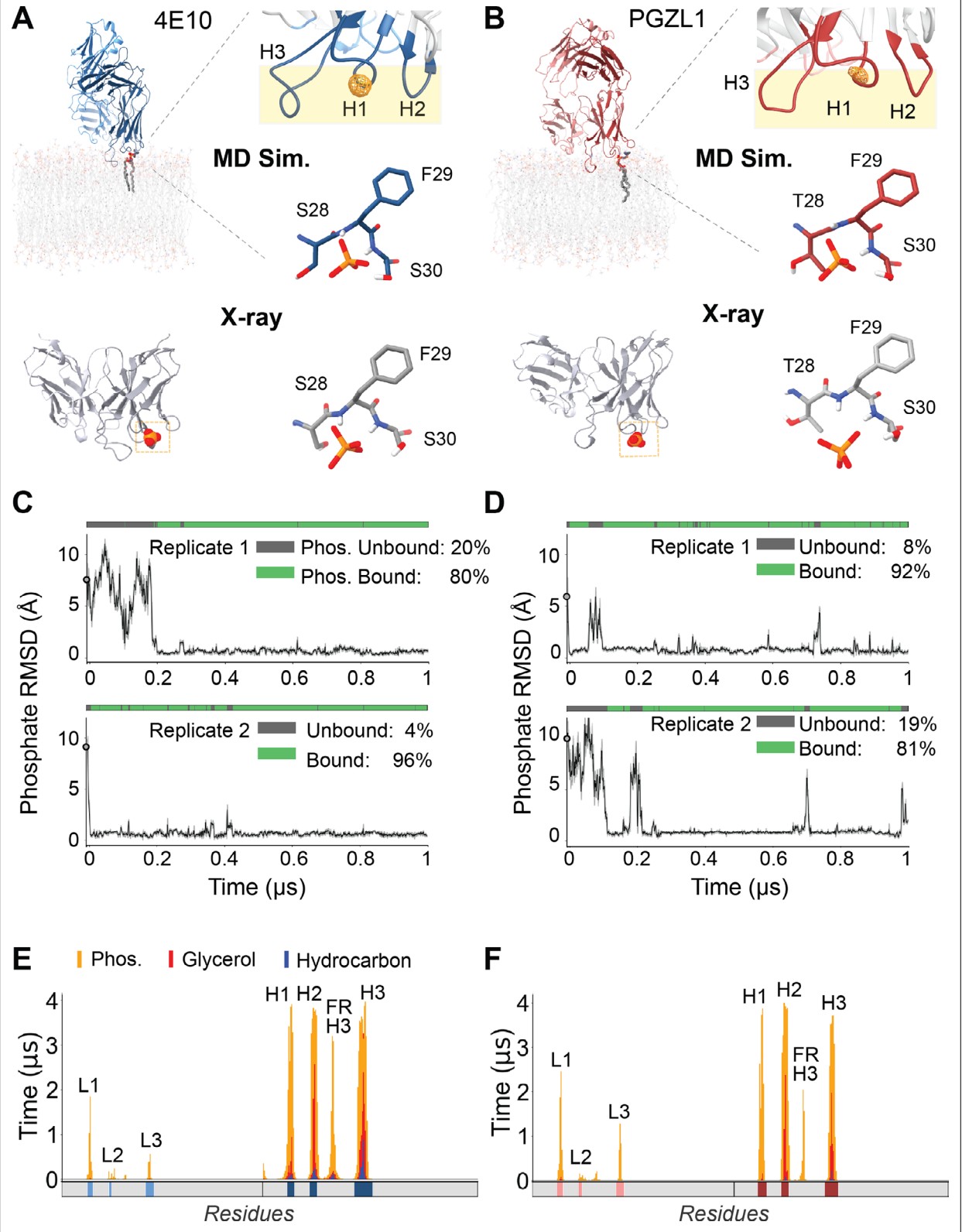

**Figure 1.** 4E10 and PGZL1 CDR lipid phosphate binding sites and membrane interaction profiles in all-atom fluid HIV-like bilayer MD simulations. (**A**) Representative frame from an MD simulation of the phospholipid complex at 4E10 Fab CDR-H1 (top left). Top right, time-averaged lipid phosphate density (orange mesh) relative to antibody CDR loops embedded in the bilayer (beige) from 4 μs total time (top right). Center right, CDR-H1 loop side chain and backbone atoms of de novo predicted phosphate interaction (middle right). Bottom, putative lipid phosphate binding site observed in

*Figure 1 continued on next page*

*Figure 1 continued*

an X-ray structure for 4E10 (PDB: 4XC1), comparing the CDR-H1 loop interactions to MD (bottom right). (**B**) PGZL1 lipid phosphate interaction at the CDR-H1 loop from MD simulation versus X-ray crystallography (PDB: 6O3J), demarked as in (**A**). (**C**) RMSD of the interacting lipid phosphate versus the experimental CDR-phosphate position (X-ray site), classified as bound (green) and unbound (grey) by loop-phosphate contacts (see Methods) in 2, 4E10 representative replicate trajectories. Black line, ten-frame RMSD running average; standard deviation, grey shading. (**D**) RMSD of lipid phosphate binding to PGZL1 CDR-H1 in MD simulations versus X-ray site. Phosphate binding is mapped above each MD trajectory as in (**C**). (**E**) Per-residue interaction profiles for Fab simulations of 4E10 detailing the time spent for each residue in in phosphate layer (orange), glycerol layer (red), or hydrocarbon layer (blue) across aggregate 4 μs from four simulations. CDR loops are mapped in solid color blocks below each profile. Fab domain regions making significant contact are labeled, including CDRs and heavy chain framework region 3 (FR-H3). (**F**) Per-residue interaction profiles for antibody Fab simulations for PGZL1, colored as in (**E**).

The online version of this article includes the following figure supplement(s) for figure 1:

**Figure supplement 1.** All atom MD replicates for 4E10 and PGZL1 with phosphate group interactions.

**Figure supplement 2.** Detailed per-residue protein-lipid interaction analysis aggregated across simulations including primary sequence to fine 4E10 and PGZL1 bnAbs membrane-interacting region.

(ns) from lipids diffusing in from ≥ 8 Å away (*Figure 1C, D*). In some cases, phospholipids bound as early as the protein-restrained equilibration (15 ns) where lipids can freely diffuse (*Figure 1—figure supplement 1D, E*). For both 4E10 and PGZL1, one of each replicate simulation had delayed formation of the stable CDR-H1-mediated phospholipid complexes (within 200 ns) requiring minor Fab reorientation on the membrane surface given these replicates' initialization from an artificially tilted orientation (*Video 1*). Upon formation, most phospholipid complexes were highly stable, typically persisting for hundreds of nanoseconds with low mobility of the phosphate headgroup relative to CDR-H1 (<1 Å RMSF) and high overall occupancy (both >80% across 4 μs total). An extensive polar network coordinates the lipid phosphate oxygens within the CDR-H1 loop: hydrogen bonds donated from backbone amides of Phe29 and Ser30 and from sidechain hydroxyls of Ser29/Thr29 and Ser30 (*Figure 1C, D*; *Zhang et al., 2019*; *Irimia et al., 2016*). Notably, CDR-H loops in contact with the membrane maintained internally rigid backbone conformations during simulation with CDR-H1 having <1 Å RMSF (*Figure 1—figure supplement 1C, D*). One PGZL1 simulation captured a rare dynamic lipid exchange event (*Figure 1—figure supplement 1E*; *Video 2*). A POPC molecule, bound for ~500 ns, dissociated from the CDR-H1 and was promptly replaced by a second nearby lipid, POPA. These observations, including the kinetic stability and high occupancy, suggest that 4E10 and PGZL1 family antibodies likely are predominantly bound to an anchoring phospholipid when intimately contacting the membrane surface, including during MPER engagement and in the neutralized complex. When CDR-H3 is inserted, CDR-H1 is presented as a rigid pre-formed binding site conveniently at the membrane's headgroup layer for rapid formation of complexes, with those phospholipids likely in a steady-state equilibrium freely exchanging (nano-microsecond timescale) yet heavily favoring the bound state.

These results of reliably recovering experimentally determined CDR-phospholipid complexes bolsters confidence that antibody behaviors within the simulations are biologically relevant,

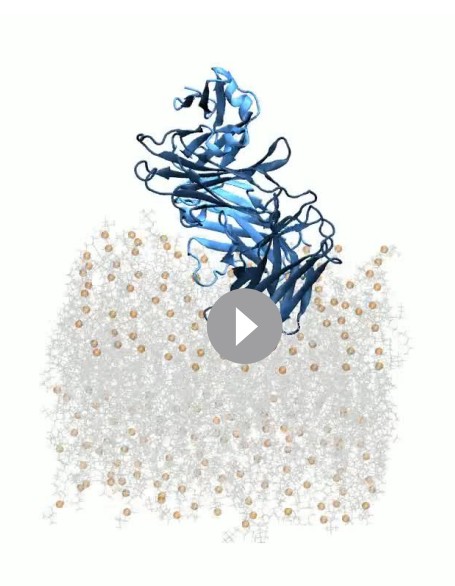

**Video 1.** De novo predicted phosphate binding in atomistic 4E10 Fab MD simulation Atomistic simulation of 4E10 Fab (heavy chain: dark blue, light chain: light blue) initially docked to the membrane using OPM PPM server prediction. Lipids and cholesterol are shown as grey sticks with phosphates from top and bottom leaflets shown as orange spheres. The binding phosphate (large orange sphere) from a POPC lipid is initially more than 10 Å away from CDR-H1 loop and finds interaction with CDR-H1 residues in first 200 ns of simulation and maintains for remainder of 1 μs.
https://elifesciences.org/articles/90139/figures#video1

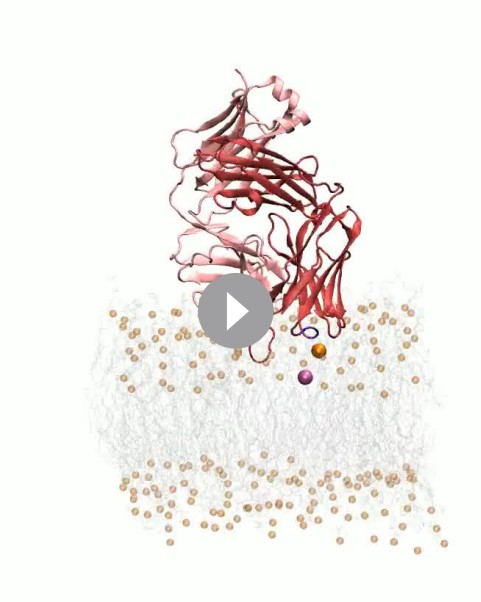

**Video 2.** Phosphate binding and replacement event in PGZL1 bnAb Fab all-atom simulation Atomistic simulation of PGZL1 Fab (heavy chain: salmon, light chain: light pink) initially docked to the membrane using OPM PPM server prediction. Lipids and cholesterol are shown as grey sticks with phosphates from top and bottom leaflets shown as orange spheres. The first interacting phosphate from a POPC lipid (large orange sphere) is initially more than 6 Å away from CDR-H1 loop and binds in CDR-H1 loop for first 500 ns of simulation. A second phosphate from a POPA lipid (pink sphere) replaces initial interaction around 550 ns and maintains the CDR-H1 interactions for the rest of the microsecond simulation.

https://elifesciences.org/articles/90139/figures#video2

motivating our deeper analysis of features driving the bnAbs' protein-lipid interactions and global membrane surface conformations. Characterizing the cumulative per-residue interaction profiles of 4E10 and PGZL1 with bilayer lipids categorized into headgroup, glycerol, or hydrocarbon layers illustrates the deep immersion of heavy-chain loops, with the light-chain making some peripheral polar headgroup contacts (namely CDRs L1 and L3; *Figure 1E, F*). The patterns of lipid interactions compare well with previous 4E10 simulations (*Carravilla et al., 2020a*; *Carravilla et al., 2020b*), yet are more expansive here – possibly due to the sustained phospholipid-CDR-H1 complex and greater simulation length.

Aligning interaction profiles with corresponding primary sequences contextualizes the amino acid molecular features responsible and their origin in antibody maturation. One novel observation consistent between our 4E10 and PGZL1 simulations was that heavy-chain framework region 3 (FR-H3) extensively intercalates into the bilayer surface. This finding demonstrates that predominantly germline encoded antibody surface features can provide significant contributions to the membrane binding mechanism and surface orientation of antibodies – a key insight for invoking and honing antibody-lipid interactions in vaccine design. For the light-chain loops and FR-H3 of PGZL1 and 4E10, charged residues mediate their shallower headgroup surface interactions (*Figure 1—figure supplement 2A, B*), whereas the CDR-H loops bare neighboring stretches of hydrophobic and lipophilic polar residues facilitating their deeper embedding. Compared to PGZL1, 4E10's CDR-H loops are further buried, more extensively contacting lipid

aliphatic tails – which may in part explain 4E10's higher tendency for poly-specificity. The comprehensive protein-lipid atomic-detail maps further detail that the mechanisms for membrane interaction and antibody maturation driven are largely by the solvation for apolar surface loops – the extent of hydrophobic burial – complemented by compatible polar residue electrostatics. The majority of bnAb protein-lipid interactions arise from surface residues' macroscopic chemical properties, contrasting the much more sequence and conformationally specific phospholipid binding sites constructed at CDR-H1. Thus, the multivalent combination of surface features combined with highly specific lipid anchoring sites may be a convergent theme for maturation and mechanism for MPER-targeting bnAbs, including the previously undervalued contribution from germline-encoded framework regions to lipid association.

## Phospholipid complex formed in the CDR Light Chain groove of 10E8 bound to membrane surfaces

We similarly investigated bnAb 10E8, which differs in its genetic origins and expected light-chain-mediated membrane binding mode (*Irimia et al., 2016*; *Soto et al., 2016*; *Georgiev et al., 2014*; *Williams et al., 2017*). In all four replicate simulations, we observed a POPC complexed at a groove between CDR-L1 and FR-L3 (*Figure 2A, B*) which had had significant non-protein electron density in previous 10E8 X-ray structures with and without lipids, which had been modelled as headgroup

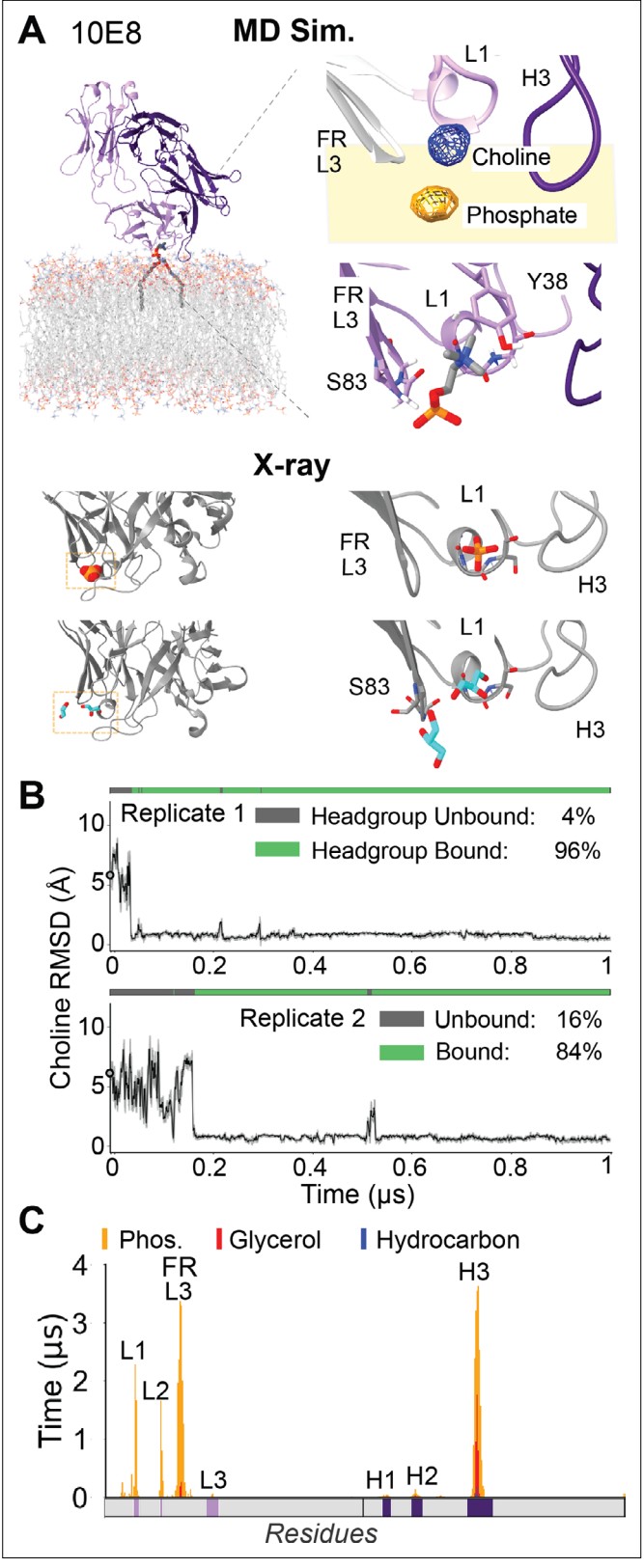

**Figure 2.** 10E8 bivalent lipid headgroup interaction and bilayer insertion predicted in MD simulationagrees with proposed crystallographic lipid binding pocket. (**A**) Top left, representative frame from MD simulation of lipid interacting with 10E8 Fab. Top right, the MD lipid binding site includes a bivalent choline and phosphate lipid headgroup complex (represented as blue and orange mesh time-averaged positional density in simulations)

*Figure 2 continued on next page*

*Figure 2 continued*

within a protein surface groove composed of CDR-L1 and FR-L3, respectively. Bottom, positions of phosphates or glycerols modeled at the CDR-L1 and FR-L3 groove site within 10E8 Fab X-ray structures (PDB: 5T85, 5T6L). (**B**) RMSD of lipid choline position in the MD simulations versus expected CDR-L1 lipid binding site from 10E8 X-ray structures. 'Bound' state assigned relative to choline position and phosphate FR-L3 interactions observed in MD. (**C**) Per-residue interaction profiles for antibody Fab simulations for 10E8 with Fab domain regions making significant contact labeled, including CDRs and light chain framework region 3 (FR-L3).

The online version of this article includes the following source data and figure supplement(s) for figure 2:

**Figure supplement 1.** All atom MD replicates for 10E8 detailing molecular interactions in phospholipid binding and membrane association.

**Figure supplement 1—source data 1.** Statistics of each metric used to define geometric macro-substates in global clustering which included aggreagating all simulation time from 10E8, PGZL1, and 4E10 together (*Figure 2—figure supplement 1A–C*).

**Figure supplement 2.** Atomic simulations robustly capture phospholipid binding within predicted sites in realistic HIV-like membranes.

**Figure supplement 3.** Structural database mining shows bnAb CDR phosphate sites from MD simulations are native-like and present in the proteome.

**Figure supplement 3—source data 1.** Bioinformatics descriptions for ligand bound structural matches and benchmarking Loop Conformations.

**Figure supplement 4.** Aggregated and independent geometric clustering of bnAb surface-bound conformations.

phosphoglycerol anions, glycerol, or free dihydrogen phosphate (*Irimia et al., 2017*). In our simulations, the stably bound POPC phosphate was slightly offset (2.7 Å) from the proposed CDR-L1 crystallographic site (*Figure 2—figure supplement 1B*), and instead localized to FR-L3 stabilized by a hydrogen bonding network from Ser83 hydroxyl, Gly84 backbone amide, and Asn85 C-alpha proton (*Figure 2A*). Simultaneously, the POPC choline moiety occupied the precise crystallographic CDR-L1 site, wherein the choline cation was coordinated by four pre-organized unpaired backbone carbonyls stemming from the specific CDR-L1 short loop helix conformation (*Figure 2A*). We find these bivalent complexes engaging both moieties of the lipid phosphocholine (PC) zwitterion at this CDR-L1 FR-L3 groove with >70% occupancy overall across replicate simulations with high stability: 0.5 Å RMSF for CDR-L1–choline site and 0.5 Å RMSF for FR-L3–phosphate site (*Figure 2—figure supplement 1C, D*). These 10E8 loops were conformationally rigid (RMSF <1.0 Å, *Figure 2—figure supplement 1E*), presenting a pre-formed binding site analogous to CDR-H1 in 4E10 and PGZL1. Previous experiments highlight the importance of CDR-L1 phospholipid binding site features in 10E8's neutralization mechanism, given that the double mutant of R29 and Y38 to alanine exhibited vastly decreased neutralization (>500-fold) across HIV strains with negligible impact to MPER affinity (*Irimia et al., 2017*). The time-averaged per-residue lipid interaction profile demonstrates how 10E8 uses both heavy (CDR-H3) and light chain loops (CDRs L1, L2, L3; FR-L3) to orient the antibody geometry on the membrane surface to position 10E8's phospholipid binding groove site at the headgroup-rich bilayer region (*Figure 2—figure supplement 1F*). These simulations supplement experimental findings with novel atomic details to establish that genetically distinct MPER antibodies share a consistent mechanism of utilizing CDR loops and framework to mediate bilayer interactions, with 10E8 utilizing FR-L3 for phosphate specificity in a stable lipid complex to anchor this bnAb at the membrane surface.

## Phospholipid binding modes in complex high cholesterol lipid bilayers

We next evaluated whether such bnAb conformations and ab initio phospholipid binding are robust to distinct lipid compositions in more complex high-cholesterol bilayers matching lipidomics studies for HIV particles (*Alam et al., 2007*; *Brügger et al., 2006*). New simulations were initialized for similar 4E10, PGZL1, and 10E8 Fab conformations in membranes having a 9.5:19:8.5:18:45 ratio of POPC, 1-palmitoyl-2-oleoyl-sn-glycero-3-phosphoethanolamine (POPE), 1-palmitoyl-2-oleoyl-sn-glycero-3-phospho-L-serine (POPS), palmitoyl sphingomyelin (PSM), Cholesterol. In a 500 ns trajectory for each, stable phospholipid complexes rapidly formed within the respective CDR loop sides with atomic accuracy to the expected position – now having more diversity in lipid headgroup chemistries (*Figure 2—figure supplement 1A–C*). Although 4E10 bound only POPC at its CDR-H1 site, in PGZL1

simulations CDR-H1 coordinated a PSM molecule (PC headgroup) in precisely the same binding mode as observed for POPC despite PSM's substitution of two lipid tails off the glycerol backbone. Likewise, 10E8 bound a POPE at its light chain groove site, with the PE (phosphate and cation) forming analogous interactions (<1 Å RMSD to POPC), including coordination of the cationic ethanolamine by the ring of backbone carbonyls capping CDR-L1's loop helix. Thus, these bnAb phospholipid binding sites can accommodate a broader scope of lipid tail and headgroup chemistries possibly beyond those we have investigated here, and can readily form stable phospholipid anchoring complexes in bilayers of HIV-like cholesterol-rich composition. One shared conformational difference observed for these bnAbs the higher cholesterol bilayers was slightly more extensive and broader interaction profiles as well as modestly deeper embedding of the relevant CDR and framework surfaces loops (*Figure 2—figure supplement 1D–F*). These results bolster the feasibility for using all-atom MD as an in silico platform to explore differential phospholipid affinity at these sites (i.e. specificity studies) and influence on antibody preferred conformation as membrane composition and lipid chemistry are systematically varied. More thorough simulation techniques such as potential of mean force are needed to make those predictions and could better guide experiments concerning how to incorporate the antibody-lipid interaction in neutralization and maturation of bnAbs via vaccine.

## Structural bioinformatics contextualizes bnAb interactions at the membrane

We sought to interrogate the biological relevance and in silico predictability of these observed CDR-lipid polar interactions using an orthogonal bioinformatics approach. We mined protein-ligand interactions to assess whether phosphate complexes of similar geometries have been observed in Nature, positing that the occurrence of analogous phosphate interactions outside the context of MPER bnAbs, that is within critical structural or functional regions of other protein families, can differentiate common proteome-wide functional motifs from simulation artifacts. Structural searches querying the lipid-binding loop backbone conformation of each Fab (as previously described *Cardoso et al., 2007*; *Zhang et al., 2019*; *Irimia et al., 2017*) identified between $10^5$ to $2 \cdot 10^6$ geometrically similar subsegments within natural proteins (<2 Å RMSD) (*Polizzi and DeGrado, 2020*), reflecting they are relatively prevalent (not rare) in the protein universe, comparing well with frequency of other surface loops of similar length in antibodies (*Figure 2—figure supplement 3—source data 1*). The structurally similar loops found in distinct proteins were subsequently mined for nearby phosphate/phosphoryl and sulfate/sulfo ligands (*Figure 2—figure supplement 3*; *Figure 2—figure supplement 3—source data 1*). When searching for loops similar to the CDR-H1 of 4E10 and PGZL1, only four cases of phosphate-type ligand binding motifs were identified. Thus, this CDR-H1 site is a realistic but rare protein-phospho-ligand structural motif (*Figure 2—figure supplement 3A, B*).

For 10E8's CDR-L1 site, only 1 phosphate-type ligand was observed: a solvent-exposed surface crystallographic ion. By contrast, 10E8's FR-L3 beta-turn site was a hot spot for phosphate-type ligands, with 23 natural protein cases (*Figure 2—figure supplement 3C*). Thus, while both CDR-L1 and FR-L3 10E8 loops sites can bind phosphate ligands, the much greater natural prevalence of phosphates ligated at FR-L3 supports that the newly proposed 10E8 POPC complex observed in our simulations is biologically realistic, rather than a simulation artifact. Barring experimental validation, existence of this FR-L3 site establishes precedent for how important framework regions can be for mediating protein-membrane interactions, including hosting stable and specific phospholipid complexes. This bioinformatics analysis illuminates the sophisticated molecular recognition mechanism in which 4E10, PGZL1, and 10E8 Fab domains incorporate recurring structural features from Nature to coordinate specific phospholipid binding interactions.

## Global geometries of membrane-bound bnAb and implications for tilted Env engagement

Given the emphasis on antibody tilt relative to the bilayer and antigen in discerning neutralization mechanisms (*Rantalainen et al., 2020*), we compared the geometries of membrane-bound conformational ensembles for our 4E10, PGZL1, and 10E8 simulations to those of Fab-bound Env trimer cryo-EM structures in membrane mimics (*Rantalainen et al., 2020*). Simulated antigen-free Fabs favor similar orientations on lipid bilayer surfaces as in bnAb-bound tilted Env experimental structures, likely facilitated by the anchoring phospholipid complexes (*Figure 1—figure supplement 1A*; *Figure 2—figure*

*supplement 1A*). The primary difference is Env-bound Fabs in cryo-EM adopt slightly more shallow approach angles (~15°) relative to the bilayer normal. The simulated bnAbs in isolation prefer slightly more upright orientations, but present CDRs at approximately the same depth and orientation. Thus, these bnAbs appear pre-disposed in their membrane surface conformations, needing only a minor tilt to form the membrane-antibody-antigen neutralization complex. The apparent barrier for re-orientation is likely much less energetically constraining than shielding glycans and accessibility of MPER. Recent studies documenting Env's inherent stalk flexibility and wide range of ectodomain tilts also suggests looser steric restriction from Env's soluble domain for an antibody's bilayer and MPER access than previously thought (*Yang et al., 2022*; *Mangala Prasad et al., 2022*).

Investigating free antibodies' membrane-bound conformational preferences, for example during maturation or for targeting privileged germlines, may be an informative approach to help illicit bnAbs via vaccine with similar lipid-dependent neutralization mechanism. Thus, we characterized the geometric range of 10E8, 4E10, and PGZL1 ensembles within our simulations via structural clustering and analyzed orientation-dependence of their CDR-phospholipid complexes. Fab surface geometries were defined by immersion depths of each CDR loop and two global domain angles relative to membrane plane and thresholded to distinguish seven substates, distributing well into population fractions from 5% to 35% (*Figure 2—figure supplement 4A–H*). Clustering of all bnAb simulation time together expectedly resulted in 4E10 and PGZL1 mapping to shared macroscopic states while the more light-chain directed 10E8 clustered separately (*Figure 2—figure supplement 4B, C*), reflecting the large differences to CDR-H1 depth and approach angle (*Figure 2—figure supplement 1—source data 1*). Upon clustering the 4E10 ensemble, interestingly, all substates maintained 86–98% occupancy of the CDR-H1 phospholipid complex, whereas 10E8 and PGZL1 ensemble substates showed more conformation-specific variance in their respective phospholipid binding (*Figure 2—figure supplement 4E, G, I*). For example, two 10E8 substates exhibit low CDR-L1 PC headgroup occupancy (23%, 33%) due to a domain rotation (20–30°) resulting in more CDR-H contacts at the expense of the more typical CDR-L interactions typically observed for 10E8 (*Figure 2—figure supplement 4I*). These findings highlight the important interplay between bnAbs' preferred surface orientation and loop composition to form these conformationally sensitive anchoring complexes through positioning of the key phospholipid-binding loops. Loop composition and conformational ensembles are likely honed together during the maturation process.

## Antigen influence on membrane bound conformations and lipid binding sites for LN01

We next studied bnAb LN01 to interrogate differences in the antibody surface-bound conformational ensemble and phospholipid interactions in the context of a transmembrane (TM) antigen, utilizing the availability of Fab crystal structures complexed with an MPER-TM fragment (*Figure 3*; *Pinto et al., 2019*). MPER-bound LN01 structures bind one phospholipid and one dodecyl-PC detergent in its CDRs, implying possible paratope-epitope-membrane cooperativity in the neutralization mechanism and two putative phospholipid sites anchoring the bnAb to membrane surfaces (*Figure 3—figure supplement 1A*). The PC headgroup binding site resides in the CDR-H3-L1-L2 inter-chain groove, with the lipid phosphate engaged by CDR-L1 Lys31 via salt bridge and the choline moiety interacting in a cation-pi cage with Tyr32 and CDR-H3 Tyr100g. When present, MPER contributes additional choline-aromatic stacking (Trp680, Tyr681). The second more solvent-exposed site lies between CDR-H1 and -H2, ordering PS or PC headgroups in crystal structures. We chose to retain the MPER-TM fragment (668-709) as a monomeric continuous helix from the LN01-bound structure (*Pinto et al., 2019*). This conformation has been observed for a similar MPER-TM peptide by NMR *Chiliveri et al., 2018* and within bnAb-bound Env by cryo-EM for TM domains in both crossing (dimeric) or separated (monomeric, 'tripod') arrangements (*Rantalainen et al., 2020*). We recognize the caveat that the model MPER-TM antigen fragment studied here does not encompass the full structural breadth of MPER display observed or hypothesized for antigen fragments or full-length Env conformational states. However, alternative MPER-TM organizations observed from peptides (e.g. trimeric TM domain helical bundle, kinked MPER in membrane surface-bound or globular trimer; *Piai et al., 2021*; *Kwon et al., 2018b*) are less amendable to model due to lack of structural data with bnAbs.

Simulations of LN01 were prepared analogously to the other bnAbs with simplified model membranes, replacing 1,2-dioleoyl-sn-glycero-3-phospho-L-serine (DOPS) in place of POPA to provide

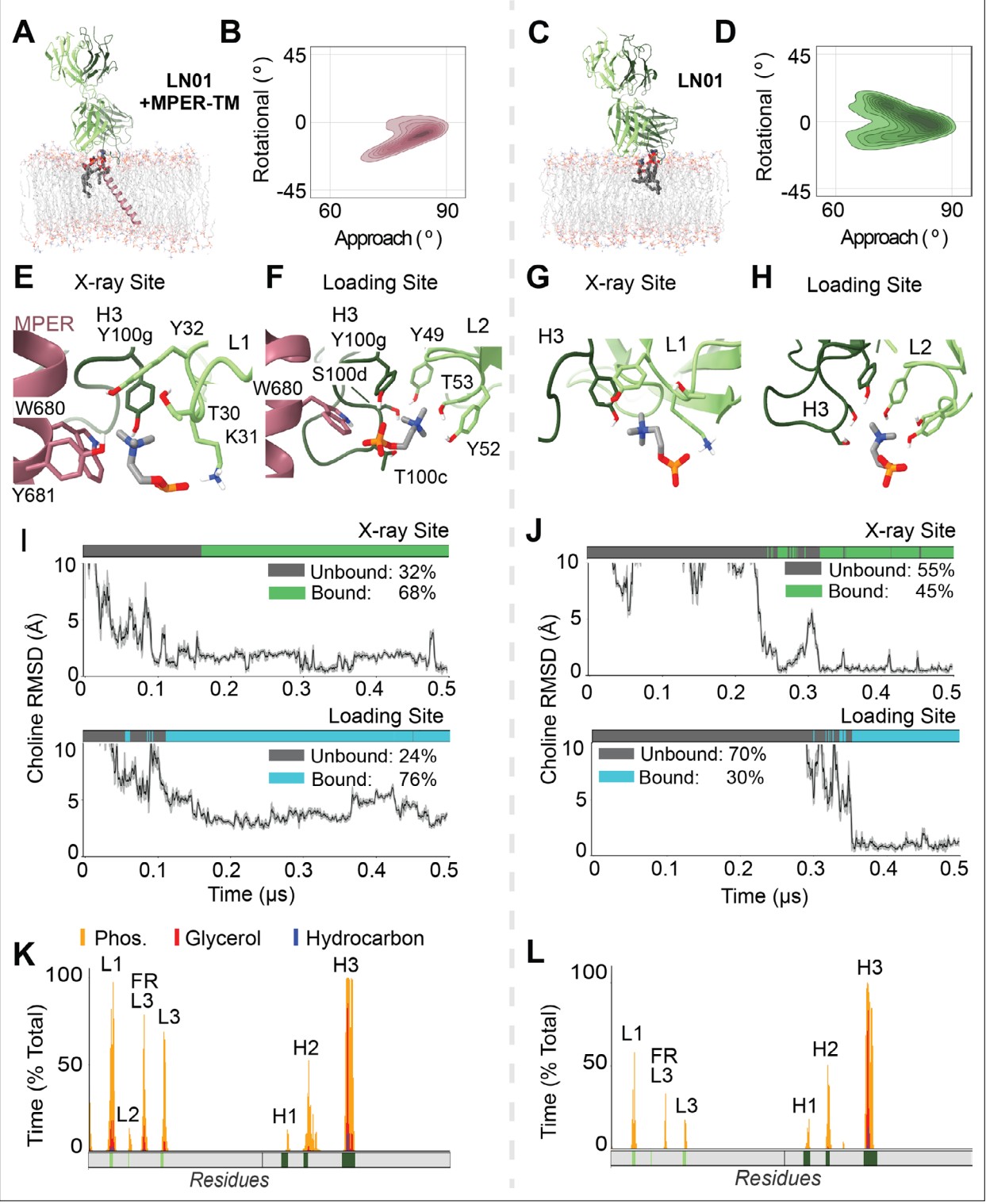

**Figure 3.** Atomistic simulations of apo and antigen-bound LN01 characterizing paratope-phospholipid complexes with and without epitope and shifted membrane-bound conformation. (**A**) Representative frame from MD simulation LN01 Fab bound to MPER-TM showing the stable ternary paratope-epitope-membrane complex; bound phospholipids shown. (**B**) Frequency of Fab's characteristic surface-bound geometry by global domain rotation and approach angles in MD simulations for LN01 bound to MPER-TM, plotted by kernel density estimation as contour. (**C**) Representative frame from MD simulation of phospholipids complexed with LN01 Fab alone. (**D**) Frequency of geometries sampled for apo membrane-bound LN01. (**E**) Phospholipid headgroup interaction formed ab initio in LN01 +MPER-TM simulations. Aromatic cation-pi cage motifs coordinate choline while the phosphate is

*Figure 3 continued on next page*

*Figure 3 continued*

coordinating by Lys31 matching the X-ray site binding pose. (**F**) The additional distal 'Loading' phospholipid site predicted in LN01simulations, with a similar cation-pi cage motif and hydrogen bonds interactions stabilizing the PC headgroup. (**G**) Atomic interactions at the X-ray site in apo LN01 simulations. (**H**) Interactions at the loading site in apo LN01 simulations. (**I**) Lipid headgroup binding in representative simulation of LN01 +MPER-TM (n=4 total). Top, X-ray binding site occupancy (green) and phospholipid choline RMSD in a representative trajectory versus experimental position. Bottom, loading site occupancy (cyan) and choline RMSD versus average headgroup bound position. (**J**) Lipid headgroup binding for representative apo LN01 trajectory (n=4 total). Site occupancy and RMSD versus predicted binding position for X-ray site (top) and loading site (bottom). (**K**) Per-residue interaction profile for MPER-TM-bound LN01 aggregated 1.5 µs from 3 simulations. CDR loops are mapped in solid color blocks below each profile. Fab domain regions making significant contact are labeled, including CDRs and light chain framework region 3 (FR-L3). (**L**) Per-residue interaction profile for apo LN01.

The online version of this article includes the following figure supplement(s) for figure 3:

**Figure supplement 1.** All atom MD replicates LN01-MPER-TM and LN01 with phosphate group interactions in X-ray and loading sites.

opportunity to reconstitute LN01's crystallographic PS binding site. Initial membrane surface geometries for apo and MPER-bound states predicted using PPM (*Pinto et al., 2019*), which accounted for solvation of the MPER-TM fragment, lead to similar embedded configurations having CDR-H3 deeply inserted with surface contacts from framework H3, light chain (CDR-L1, L3), and heavy chain (CDR-H1, L2). Across three 1 µs replicates each, those loops remained inserted and adopt ensembles similar to the starting conformations (*Figure 3A, B, K, L*). The MPER-antibody interface including key interactions (e.g. Trp100H to Thr676 H-bond) remained stable. The TM domain maintains a continuous helix through MPER without kinking in a stable tilted geometry (minimal rotation about helix axis) that enables the previously observed snorkeling and hydration of TM Arg686 (*Hollingsworth et al., 2018*). In both apo and MPER-TM-bound replicates, several POPC binding events were observed within the CDR-H3-L1-L2 groove at atomic accuracy when compared to dodecyl-PC headgroup position in LN01 X-ray structures, with the same sidechain interactions (*Figure 3E–J*). In contrast to PGZL1, 4E10, and 10E8 simulations, phospholipids in LN01's CDR-H3-L1-L2 groove site have much lower kinetic stability and occupancy: ~30% and~40% aggregate occupancy for MPER-TM-bound and apo, respectively (*Figure 3—figure supplement 1C, D*). Occupancy at the second crystallographic CDR-H1-H2 PS/PC site was negligible within any trajectory. Thus, unbiased MD identifies the CDR-H3-L1-L2 groove as the primary phospholipid anchoring site for LN01 in biologically realistic membranes both alone and when bound to TM-embedded MPER, and further emphasizes the importance of lipid in LN01's maturation and neutralization mechanism.

Two interesting behaviors stood out in LN01 simulations. First, an additional new CDR-phospholipid binding site was observed reproducibly across apo and MPER-TM-bound trajectories at an alternative groove comprised of and CDR-H2, CDR-H3, and FR-L2 we termed the Loading Site (*Figure 3E–J*). Phospholipids appeared to readily exchange the ~7 Å distance between this Loading Site and the primary 'X-ray' CDR-H3-L1-L2 groove phospholipid binding site (*Figure 3—figure supplement 1B*). Although, the two sites were often simultaneously occupied. In the Loading Site, lipid headgroup phosphate oxygens were coordinated by hydrogen bonds to LN01 Tyr100g, Tyr49, and Tyr52 – and optionally to gp41 Trp680 if present (*Figure 3E–H*). Bivalent interaction with the headgroup choline occurs simultaneously via a similar electrostatic interactions and cation-pi cage motif from Tyr49, Tyr52, Thr53, Thr100c, Ser100d, and Tyr100g sidechains. The Loading site was occupied more often than the X-ray site: 78% and 58% simulation time for MPER-TM-bound and apo LN01 replicates respectively (*Figure 3I, J*; *Figure 3—figure supplement 1C, D*).

The second notable behavior observed was that antigen-free LN01 sampled a broader range of geometries at the membrane surface, characterized by two angles describing the Fab relative to the membrane (*Figure 3A–D*). LN01 complexed with MPER-TM was more deeply and extensively embedded than LN01 alone and resulted in a more focused conformation landscape – aligned with the minimal dynamics of the MPER-TM orientation. Protein-lipid interaction profiles reflect consistency in the protein features mediating membrane association between apo and MPER-TM-bound trajectories (*Figure 3K, L*), with CDR-L and FR-L3 more inserted upon TM antigen engagement upon the minor rearrangement to engage transmembrane antigen. Critically, while apo LN01 favors a different geometry (positive Rotation angle) its membrane surface ensemble heavily samples the major conformation adopted in MPER-TM-bound state (negative Rotation angle; *Figure 3B, D*). This observation that LN01's surface-bound geometry is predisposed for transmembrane antigen engagement

suggests that maturing bnAbs' sequences and structures are primed for the membrane interaction key to access MPER and form the neuralization complex.

## Coarse-grain ab initio insertion simulations capture biologically relevant membrane-bound conformations

Next, we used extended sampling time afforded by Martini coarse-grained (CG) simulations to more thoroughly explore the process of antibody insertion to lipid bilayers and landscape of feasible bnAb surface-bound conformations (*Corey et al., 2019*). Fab's simulated in CG representation with elastic network restraints maintained the protein fold (<2 Å backbone RMSD) and retained key heavy-light inter-domain contacts, informing that this polar bead model rigidly presents the Fab macroscopic surface features critical for mediating bilayer surface interactions, that is hydrophobic patches, charge, polarity (*Figure 5—figure supplement 1A*). Thus, CG simulations might be useful for evaluating bnAbs' propensity for membrane insertion and approximate preferred conformations, a hypothesis we probed by two distinct simulation approaches.

The first 'spontaneous insertion' approach placed a Fab at distinct random initial orientations and distances (0.5–2 nm) above a pre-assembled HIV-like anionic lipid bilayer, allowing Fabs free diffusion for 14 µs (*Figure 4A*). In 18 replicate simulations for each of 4E10, PGZL1, and 10E8, we observed CDR-directed association stable within the lipid bilayer (>1 µs) within 10, 14, and 12 trajectories, respectively (*Figure 4*, *Figure 4—source data 1*, *Video 3*). Simulations captured bilayer surface scanning behavior preceding insertion and numerous dissociation events, pointing to the reversibility and dynamics of the process. As a reference, bovine serum albumin (BSA) was tested similarly. BSA's documented weak affinity for lipid bilayer association (low millimolar; *Ruggeri et al., 2013*) aligns with the drastically reduced membrane contact events (2/18) and lack of sustained insertion observed (*Figure 4E*). Non-neutralizing anti-gp41 antibody 13h11 having no detectable lipid bilayer interaction by bilayer interferometry (*Alam et al., 2009*; *Nicely et al., 2010*) also exhibited negligible insertion and only sparse 'scanning' events, starkly contrasting the bnAbs' behaviors. Thus, these CG simulations readily distinguish MPER bnAbs with high propensity for lipid interaction (low micromolar affinity) from non-specific and low affinity lipid interactions.

A second 'co-assembly' CG approach (*Scott et al., 2008*) aimed to enhance sampling bnAbs insertion and conformations through spontaneous assembly a lipid bilayer around a randomly oriented Fab in a simulation box amongst mixed water, ion, and lipid particles (*Figure 4F*). Predisposition for Fab-lipid interactions and insertion events were increased, particularly for 10E8 and PGZL1, while time wasted sampling Fab in bulk water was reduced. We combated potential biases of this method for possibly enriching insertion in less favorable conformations and skewing distributions by running more replicates (n=40) of shorter 5 µs simulations. Most trajectories having Fabs interacting with lipid with surface features outside CDRs dissociated quickly or never inserted (*Figure 4G–I*).

We compared the spectrum of membrane-bound conformations sampled from ab initio CG membrane insertion and with pre-inserted all-atom simulations for each bnAb to characterize correlation. Relative Fab orientation was tracked via two global domain angles: the canonical 'angle of approach' (Fab long axis intersecting the membrane normal vector) and an internal 'rotation angle' from the vector orthogonal to this long axis (*Figure 5A*). Fab conformational ensembles and protein-lipid interaction patterns from both CG approaches heavily overlapped with the primary geometry observed for each respective bnAb all-atom trajectories, showing consistency between the two simulation models (*Figure 5B*, *Figure 5—figure supplement 1B, C*). CG 4E10 insertions across both methods adopted single conformations which were similar and all-atom-like. PGZL1 and 10E8 showed more promiscuity by sampling alternative ab initio membrane-inserted geometries alongside their respective all-atom-like conformations, with these states appearing consistently in both CG approaches but at different frequencies. The apparent deeper energy well for 4E10's primary geometry in CG simulations could reflect greater conformational specificity, but could also be a result of the Martini model's overestimation of hydrophobic interactions (*Srinivasan et al., 2021*).

Overall, these CG simulations facilitate study of *ab initio* membrane insertion for MPER bnAbs, wherein we find all sustained events are CDR-directed, ruling out contributions from other Fab surface features. For each bnAb, a state we deem to be biologically relevant, based on similarity to geometries observed in all-atom trajectories forming key phospholipid complexes, was either the primary conformation or one of the top 2 sampled. Given that non-CDR directed and alternative CDR-embedded

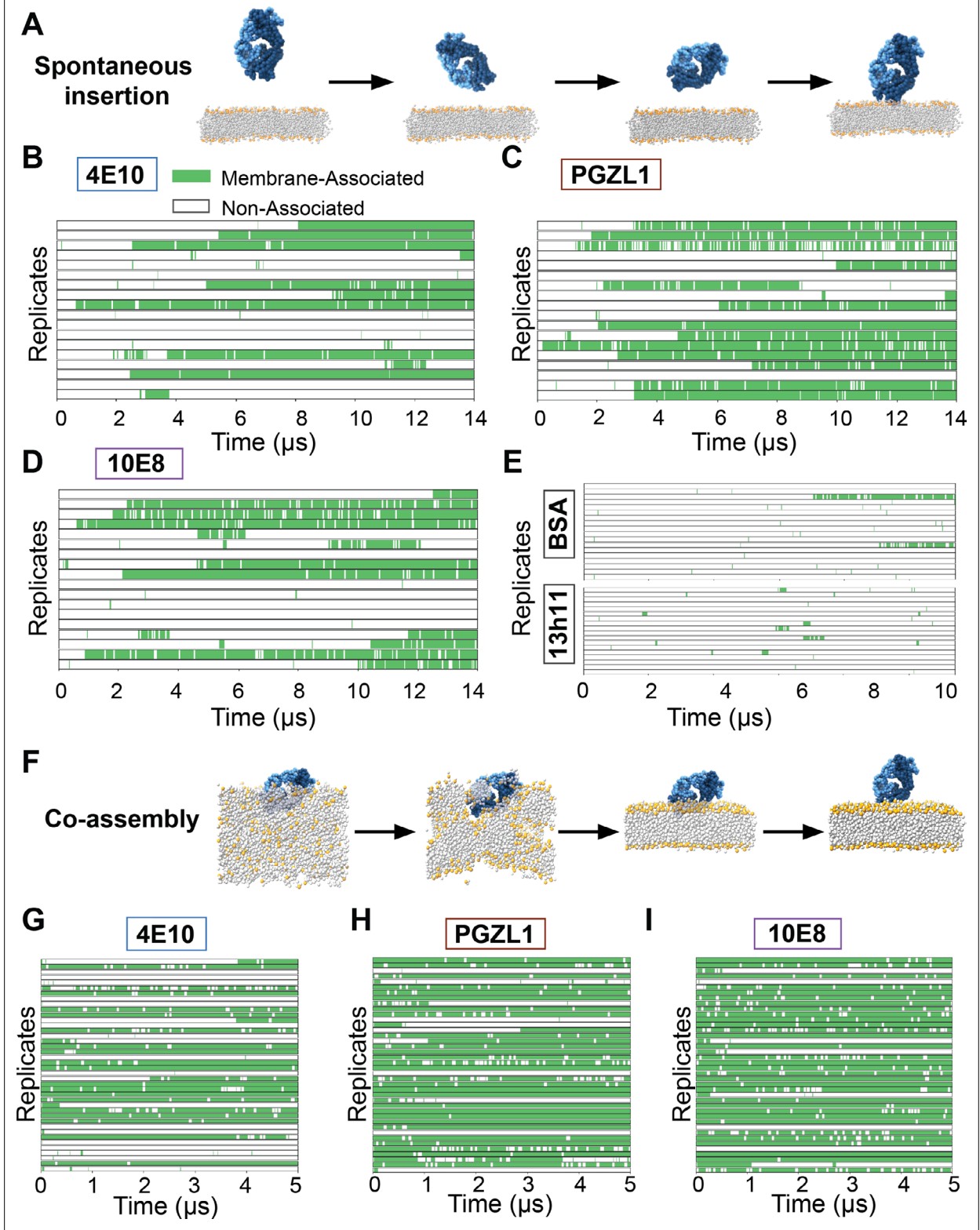

**Figure 4.** Unbiased spontaneous membrane insertion events and semi-biased dissociation events in coarse grain MD simulations. (**A**) Snapshots of spontaneous insertion event from a Martini model coarse-grain simulation of a 4E10 Fab. The Fab begins in explicit water solvent 0.5–2 nm above a lipid bilayer, freely diffusing and tumbling in bulk solvent, often resulting in a temporary or permanent insertion event (right). (**B–D**) 18 replicates of coarse grain Fab systems (4E10, PGZL1, 10E8, respectively), initialized with slightly different Fab orientations relative to lipid bilayer. Frames with Fab contacting the membrane are in green and frames with Fab in water (non-associated) are in white for replicate trajectories of 14 μs each. (**E**) 18 replicates of coarse

*Figure 4 continued on next page*

*Figure 4 continued*

grain BSA (top) or 13h11 (bottom) with different starting orientations relative to the lipid bilayer. Membrane contact (green) or diffusion in water (white) shown over 10 μs time. (**F**) Snapshots describing a co-assembling membrane pipeline with 4E10 Fab. An Fab is centered in a box with various rotational orientations in space, explicit water, and lipids randomly arranged within a subset of the box (left). By 30ns, the membrane is fully formed (middle). Fab molecules result in a pre-docked membrane bound conformation and sample a permanent insertion event, intermittent membrane association, or dissociation depending on how the Fab contacts with the membrane (right). (**G–I**) 40 replicates of 5 μs simulations for coarse grain co-assembling systems (4E10, PGZL1, 10E8, respectively), each with slightly different Fab initial orientations relative to lipid bilayer. Membrane contact is classified as above.

The online version of this article includes the following source data for figure 4:

**Source data 1.** Spontaneously inserted coarse grain membrane contact events for bnAb Fabs.

orientations readily dissociate, we conclude that course-grained models can distinguish unfavorable and favorable membrane-bound conformations to an extent that provides utility for characterizing antibody-bilayer interaction mechanisms.

## Multi-scale simulations of ab initio formation of bnAb phospholipid complexes

We next integrated these spontaneously inserted CG conformations into all-atom trajectories to track the full *ab initio* process of bnAb association and determine the competency of CG-derived geometries to subsequently acquire the critical phospholipid anchoring complexes. For each of 4E10, PGZl1, and 10E8, three representative medoid Fab poses were extracted by clustering CG simulation frames (clusters A, B, C), and converting to all-atom detail to initiate unbiased MD simulation (*Figure 6A, B*).

For 4E10, trajectories initiated from all three geometries drifted back to conformations very similar to those of our initial pre-inserted all-atom trajectories and bound phospholipids at the CDR-H1 site (*Figure 6B–E*). Interestingly, the trajectory back-mapped from cluster A adopted deeper insertion and higher CDR-H1 phospholipid binding occupancy (>60%) compared to ensembles of clusters B and C trajectories, which were slightly tilted (rotation angle >0°) and less extensively inserted, indicating the latter geometries are likely less ideal (*Figure 6—figure supplement 1B*). For PGZL1, all-atom trajectories starting from CG clusters B and C similarly converge to CDR-H-inserted conformations analogous to our previous all-atom MD, and stably bind phospholipid headgroups at CDR-H1 (>50% occupancy; *Figure 6C–E*). For 10E8, Cluster C started and finished in membrane-bound orientations analogous to the pre-inserted simulations (*Figure 6C*), although the Fab adopted a slightly different protein-lipid interaction pattern with much shallower CDR-H3 insertion, the bivalent CDR-L1 phospholipid complex still readily formed with high occupancy (>70%) (*Figure 6D, E*). Thus, these multi-scale simulations demonstrate that unbiased CG modeling can generate stable membrane-bound conformations ab initio that integrate smoothly into atomistic simulations which are biologically relevant, in that they capture key phospholipid complexes seen in laboratory experiments inferred to be critical for bnAb function.

CG sampling also explored additional alternative membrane-bound geometries, which we assessed for kinetic stability in integrative

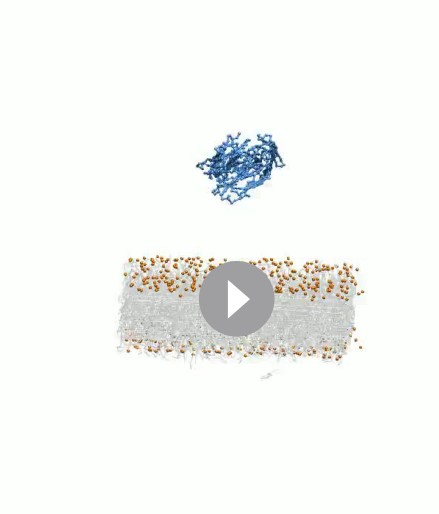

**Video 3.** Unbiased spontaneous membrane insertion event in coarse grain 4E10 Fab MD simulation Coarse grain 4E10 Fab (blue) is initialized from 2 nm above an assembled membrane (grey lipid tails and cholesterol and orange phosphates). Random diffusion and tumbling in explicit water is observed before initial membrane contact is made. Membrane association is followed by reorganization of the Fab-membrane conformation for the remainder of the 14 μs simulation.
https://elifesciences.org/articles/90139/figures#video3

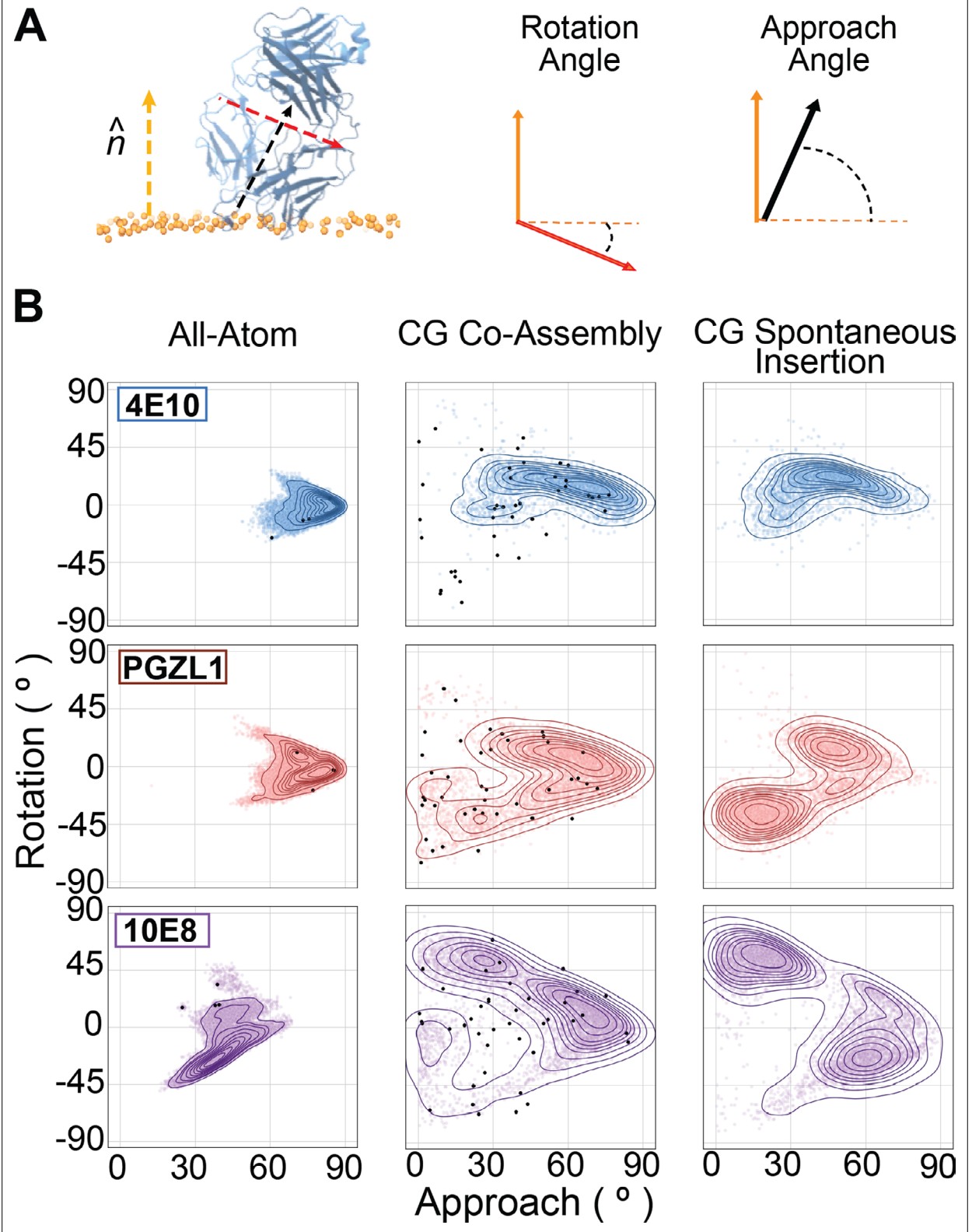

**Figure 5.** Membrane surface-bound bnAb conformations sampled across multiscale simulations. (**A**) Graphic of angles defined to describe Fab geometries relative to the normal vector at the membrane's upper leaflet lipid at the phosphate plane in simulation frames (orange arrow). The canonical 'approach angle' defines the long axis of the Fab domain (i.e. the central pseudo-symmetry axis) and membrane normal vector (black arrow). A second 'rotational angle' is defines the global domain rotation about the Fab pseudo-symmetry axis relative to the membrane normal vector, based

*Figure 5 continued on next page*

*Figure 5 continued*

on the short axis traversing the light and heavy chains, which is nearly orthogonal to the Fab's central axis (red arrow). (**B**) Frequency plots of rotation and approach angles from frames of membrane-bound Fabs in MD simulations for 4E10 (blue, top row), PGZL1 (red, middle row), and 10E8 (purple, bottom row). Contour plots depicting frequency maxima for angle pairs sampled are by kernel density estimation. Left column, membrane interaction angles sampled from all-atom simulations with Fabs pre-docked using OPM PPM server prediction. Middle column, geometries from coarse-grain membrane co-assembly simulations. Right column, geometries from unbiased spontaneous insertion coarse-grain simulations. Black dots denote the initial Fab-membrane geometries of starting states for replicate trajectories for each antibody initiated in the lipid bilayer.

The online version of this article includes the following figure supplement(s) for figure 5:

**Figure supplement 1.** Stability and membrane interactions from coarse-grain expanded sampling.

atomic simulations to discern secondary protein binding modes or intermediate conformations aiding membrane association (e.g. preliminary surface-scanning) from simple artifacts. The trajectory initiated from PGZL1's CG cluster A adopts a novel conformation stable on the 500 ns timescale with CDR-H3 inserted, favoring lipid-CDR-L interactions over CDR-H, that was not competent to complex phospholipids at CDR-H1 (*Figure 6B–D*). All-atom simulation from one alternate conformation of 10E8, CG cluster A, rapidly dissociated from the membrane within 50 ns (*Figure 6—figure supplement 1C*), showing this integrative MD distinguishes this state as an unstable likely artifact. Interestingly, the all-atom trajectory starting from CG cluster B trended towards the same novel conformation as cluster A, but remained stably inserted and bound phospholipid at the CDR-L1 site at high occupancy (70 %), albeit having shallower CDR-H3 insertion alongside canonical 10E8 CDR-L embedding (*Figure 6C, D*). These simulations hint at the feasibility to study compatibility of these lipid complex with broader Fab-membrane geometries, and reveal the need for orthogonally evaluating these conformations such as by biased simulations (e.g. potential of mean force, PMF) which we address later in this manuscript.

Additionally, we modelled the orientation of a full-length IgG based on the simulated position of each bnAb Fab to ascertain how sterically feasible a bivalent Fab-membrane interaction could be and possibility of avidity effects in mechanism. The resulting full-length IgG geometries shows that with one membrane-bound Fab, it is very unlikely a second Fabs can simultaneously engage the bilayer due to restriction from the IgG hinge – except for the case of an extraordinarily flexible hinge (*Figure 6—figure supplement 1D*).

## Biased pulling simulations distinguish experimentally characterized affinity differences

Finally, we applied biased constant velocity atomistic simulations to compute the force required to dissociate a Fab bound to lipid bilayers – providing a binding strength estimate akin to force spectroscopy (*Figure 7A*). Doing so tested whether this approach could connect the observed bnAb anchoring phospholipid complexes and membrane-bound conformations with physiological properties such as lipid bilayer affinity and neutralization potency. If informative, this method offers an expedient alternative to canonical free energy calculations such as PMFs which demand tens of μs (*Corey et al., 2019*), prohibitive for characterizing many antibody variants. Given that Fabs may sample many lipid interactions and surface conformations, we performed ensemble-based measurements: averaging forces from several replicate pulling trajectories with different starting configurations from unbiased MD.

We first assessed suitable pulling velocities through dissociating 4E10 poses from simplified anionic HIV-like lipid bilayers to a fixed 1.5 nm distance over different intervals (10, 50, 100, 200 ns; n=3 unique starting conformations). Interestingly, two of three starting poses adopt the CDR-H3 rotamer conformation of Trp100b projecting down towards lipid tails ('Trp-Down' state) while the third starting pose has Trp100b in a flipped rotamer pointing towards the lipid-water interface ('Trp-Up' state; *Figure 7—figure supplement 1A*). As expected, rupture forces decrease with increasing simulation time (slower pulling velocity), plateauing between 100 and 200 ns (*Figure 7—figure supplement 1B*). Surprisingly, the rupture forces from trajectories starting in the Trp-Down state were nearly identical (<2% difference), and rupture forces from trajectories starting from the Trp-Up state were consistently lower (27 and 47% greater) for 50 and 100 ns, respectively. This cursory trial exhibited promising conformational sensitivity in distinguishing different protein-lipid interactions in a CDR loop and precision from similar initial Fab conformations.

We next compared the behavior of 4E10 with its well-studied CDR-H3 WAWA variant, the W100aΔA-W100bΔA double mutant (*Figure 7B*). WAWA has similar $K_d$ to MPER as 4E10, but significantly reduced

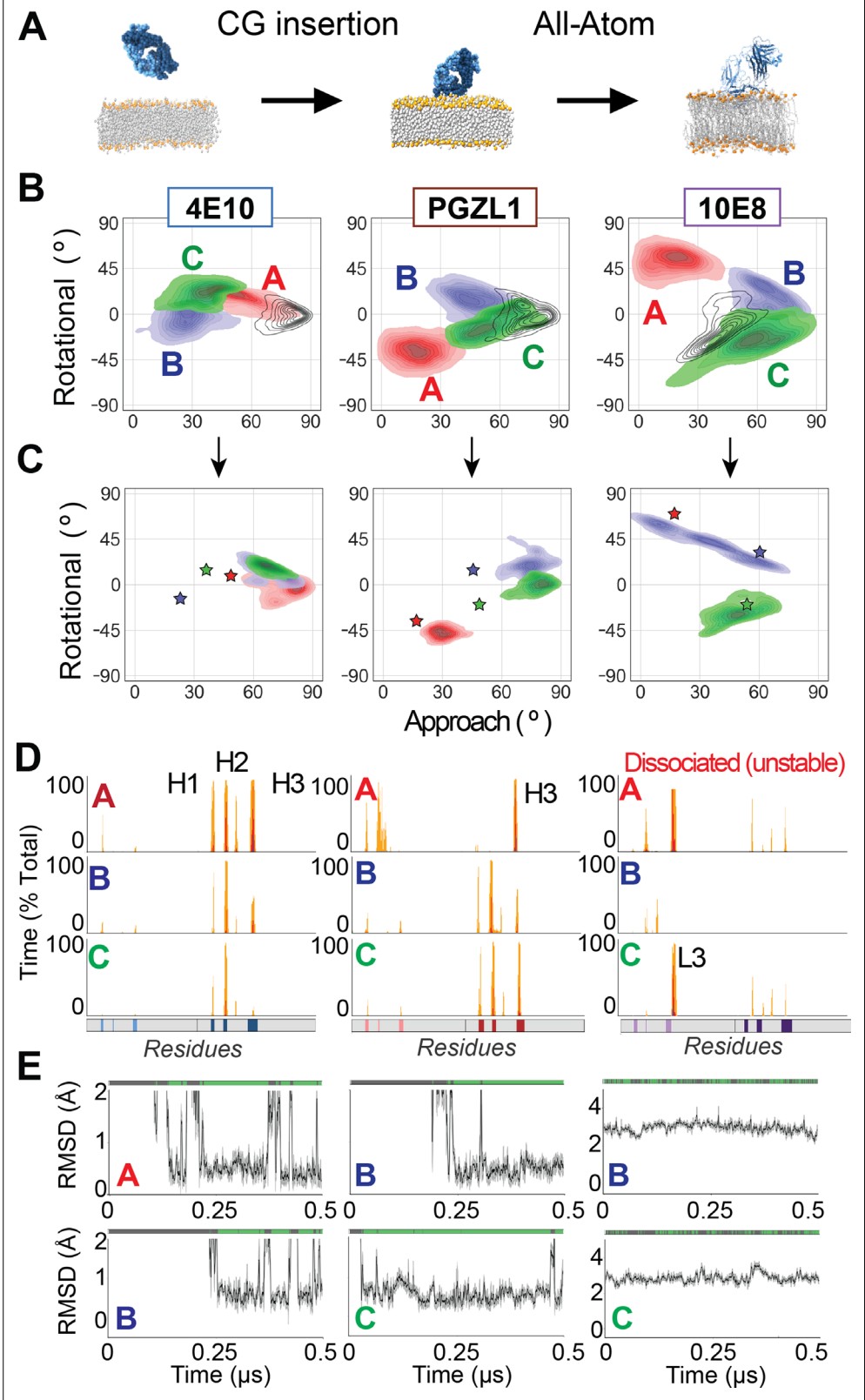

**Figure 6.** Back-mapping CG-membrane-bound geometries to all-atom simulations allows integrative ab initio modeling of the full bnAb insertion process. (**A**) Representative frames from membrane-bound 4E10 Fab coarse-grained simulations were back-mapped to all-atom representation to assess the stability and plausibility of those membrane-bound conformations. This coarse-grained-to-all-atom (CG-to-AA) reversion was applied for all medoid

*Figure 6 continued on next page*

*Figure 6 continued*

frames of interest from each antibody system and used to initiate half-microsecond unbiased all-atom dynamics simulations. (**B**) Frequency of membrane interaction angles from coarse grain spontaneous insertion as clustered by geometric substates for 4E10, PGZL1, and 10E8, colored and contoured as in *Figure 3B*. Corresponding primary all-atom simulations is overlaid (unfilled, black contour frequency density plot). (**C**) Conformational geometry sampled upon conversion of CG medoid to an all-atom trajectory. Initial geometry denoted by stars colored matching CG clusters in (**B**). Frequency and contour plots of conformational angles sampled in stably inserted backmapped all-atom trajectories for 4E10 (left), PGZL1 (middle), and 10E8 (right). (**D**) Per-residue interaction profiles for antibody Fab simulations for 4E10 (left), PGZL1 (middle), 10E8 (right) representing each backmapped atomic trajectory, showing CDR-mediated conformations of differing depths and geometries. (**E**) Phospholipid headgroup binding and RMSD plots of closest lipid at respective experimentally determined CDR sites for 4E10 (left), PGZL1 (middle), and 10E8 (right), plotted as in *Figure 1C* or *Figure 2B*. 4E10 Cluster A is reported in supplement, PGZL1 Cluster A is not reported because no lipid binding was detected, and 10E8 Cluster A is reported in supplement as an artifact.

The online version of this article includes the following figure supplement(s) for figure 6:

**Figure supplement 1.** Coarse grain to all atom backmapping validation.

affinity to empty HIV-like liposomes and neutralization efficacy (>100-fold higher IC$_{50}$; *Scherer et al., 2010*; *Kwon et al., 2018a*; *Xu et al., 2010*; *Montero et al., 2012*). Across replicates from different starting states for 4E10 (n=9) and WAWA (n=11), rupture force calculations showed average forces (± S.E.M.) of 63.8±2.8 kJ/nm/mol and 48.0±2.6 kJ/nm/mol, respectively (*Figure 7D, E*), correctly indicating the 4E10 ensemble has a stronger membrane interaction (32 ± 13%; p-value <0.001). Thus, this in silico estimation effectively discerns antibody variants whose high and low lipid bilayer affinities have been previous measured and reinforces the molecular link between WAWA's reduced membrane binding and neutralization.

Finally, we evaluated experimentally characterized PGZL1 variants that recapitulate possible stages of a germline maturation pathway (*Zhang et al., 2019*; *Figure 7C*). Mature PGZL1 has relatively high affinity to the MPER epitope peptide (K$_d$ = 10 nM) and demonstrates great breadth and potency, neutralizing 84% of a 130-strain panel. An 'Intermediate' PGZL1 with V-gene germline reversion (gVmDmJ, 100% CDR-H3 identity) showed a modest affinity loss (K$_d$ = 64 nM), while neutralization potency and breadth (12%) were more substantially reduced than expected due to affinity alone – attributed to impaired lipid bilayer interactions (*Zhang et al., 2019*). The fully 'Germline' reverted variant (gVgDgM) had no detectable MPER peptide affinity or neutralization. Ensembles for Germline and Intermediate Fab pulling simulations were prepared from unbiased all-atom trajectories, starting from predicted hydrophobicity-optimized inserted poses. The variants adopt insertion geometries like the mature Fab, facilitated by the mostly conserved CDR-H3 (*Figure 7—figure supplement 1C*), but did not form CDR-H1-phospholipid complexes. Significantly lower pulling forces were required to dissociate Intermediate and Germline PGZL1 from bilayers, 37.5±14.9 kJ/nm/mol and 41.2±11.7 kJ/nm/mol respectively, than for mature PGZL1, 56.1±11.6 kJ/nm/mol (p<0.003; *Figure 7E*). Germline PGZL1 likely reflects kinetically stable but low-affinity interactions, representing baseline of rupture forces required to dissociate CDR-inserted antibodies. While the difference in Germline and Intermediate force distributions was not significant (p>0.4), more higher force events were observed for the Intermediate. These results provide further evidence that bnAb phospholipid binding features are acquired and honed along the maturation pathway, entrenching the connection between positive selection of antibody-lipid interactions and neutralization efficacy.

## Discussion

Here, we present a roadmap for applying multi-scale molecular simulations to characterize the biophysical underpinning of lipid membrane interactions within the mechanism of HIV MPER bnAbs. The approaches demonstrated and principles discerned improve understanding of maturation pathways for antibodies targeting membrane-proximal epitopes and should enrich data-driven design of HIV immunogens. Our detailed simulations supplement in vitro binding and crystallographic evidence in establishing that bnAbs develop highly specific phospholipid interactions that facilitate access to the MPER epitope and can participate in epitope-paratope interface in context of full lipid bilayers. These results add validation to which out of the many possible ions and short-chain lipid moieties

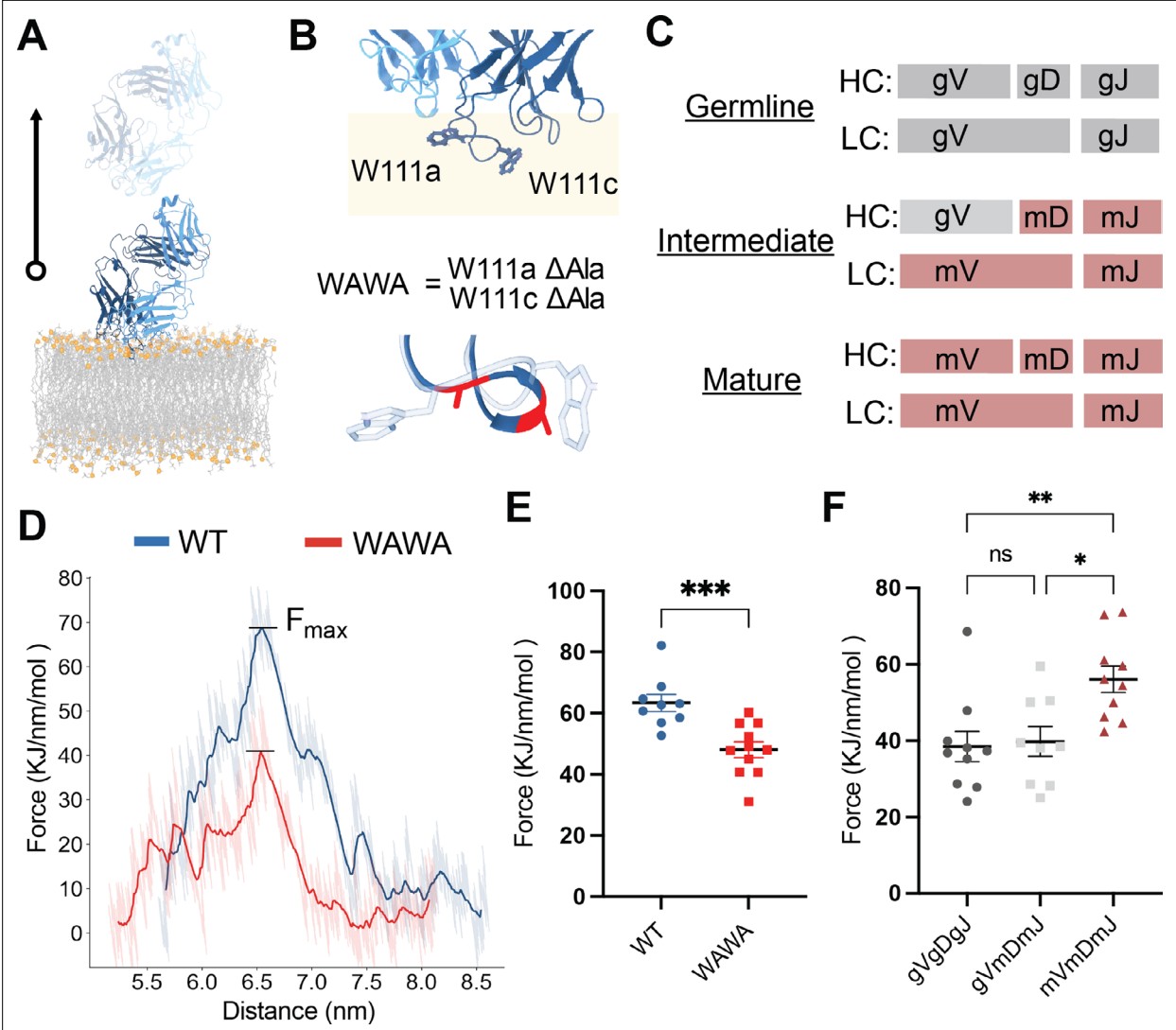

**Figure 7.** Biased antibody-membrane pulling dissociation simulations approximate Fab-membrane interaction strength for bnAb Fab variants of known lipid binding affinity or neutralization potency. (**A**) Schematic of pulling method, a bnAbs Fabs associated with the lipid bilayers is subject to an applied upward dissociation force to the Fab domain center of mass at constant velocity to measure the force required to dissociate the Fab from the bilayer. (**B**) The Trp100a-Trp100c motif residues in 4E10 CDR-H3 loop of expected to be lipid-embedded in the bilayer (beige). The double alanine mutant 'WAWA' 4E10 variant has experimentally determined significantly reduced affinity to lipid bilayers and lower neutralization potency, due to lack of Trp membrane insertion. (**C**) Previously experimentally characterized (*Huang et al., 2012*) PGZL1 germline-reverted variants, shown as chimera of germline versus matured gene segments, used to approximate antibody properties along its maturation trajectory. (**D**) Average force versus distance plots and rupture force ($F_{max}$) calculation for one replicate of pulling wild type 4E10 (WT, blue) and 4E10 WAWA (red) to bias Fab dissociation from the bilayer. (**E**) Distribution of rupture forces required for dissociation for different membrane-bound starting conformations for 4E10 (n=9, blue) and 4E10 WAWA (n=11, red), with starting conformations drawn from previous unbiased all-atom simulation ensembles at rest. Outliers as dots (**F**) Distributions of rupture forces (n=10) required for membrane dissociation of PGZL1 inferred variants along the maturation pathway, for germline (dark grey, gVgDgJ), intermediate (light grey, gVmDmJ), and mature (maroon, mVmDmJ) PGZL1. A one-tailed two-sample t-test was performed to analyze statistical significance of rupture forces between two antibodies (*=p < 0.05; **=p < 0.005; ***=p < 0.001).

The online version of this article includes the following figure supplement(s) for figure 7:

**Figure supplement 1.** Constant velocity pulling simulations are sensitive to dynamics and sequence variation.

ordered within those crystal structures may be inferred as putative bilayer phospholipids, a critical consideration in antibody and vaccine design. The additional simulations outlined with different lipid ratios and varied antigens as well as more rigorous PMF studies, could be applied to better inform affinity thresholds and energetic relationships paramount to the lipid complexes and conformational preferences reported here. Furthermore, these modeling approaches should have utility in finding

membrane-targeting elements in antibodies towards identifying causes of autoimmune stress. Likewise, the analysis tools in this work might be practical for aiding antibody engineering targeting membrane proteins, for example grafting lipid phosphate binding structural motifs or amino acid sequences with desirable membrane associating properties into distinct antibody loops regions.

Further, these simulations can serve future examination of molecular details concerning the genetic origins and developmental pathways for incipient lipid-binding antibodies. Through natural or vaccine-induced immunity, coaxing the immune system to develop both lipid and antigen affinity (possibly even cooperativity) during maturation is a difficult task, and hindered by downregulation of membrane binding precursors (*Goodnow et al., 2005*; *Verkoczy et al., 2010*; *Alam et al., 2007*; *Mascola and Haynes, 2013*). Likewise, MPER bnAbs are difficult to induce and, until recent clinical trials, have only been isolated from patients with chronic HIV-infection and sustained immunosuppression (*Rujas et al., 2024*; *Rujas et al., 2024*; *López et al., 2024*; *Stiegler et al., 2001*; *Huang et al., 2012*; *Zwick et al., 2001*). Modern vaccine strategies often elicit certain subsets of precursor B cells, often targeting specific germline genes, in attempt to guide antibodies' maturation of specific molecular features intended for the targeted epitopes (*Jardine et al., 2013*; *Steichen et al., 2016*). The question remains whether particular germline genes and lineages are privileged for successful maturation as membrane-targeting (or MPER-targeting) antibodies. This notion is supported by 4E10 and PGZL1 sharing a germline gene (VH1-69) *Zhang et al., 2019* and nearly identical membrane interactions. Certain germlines genes may be predisposed with inherent basal membrane affinity or be favored precursor scaffolds for compatibility with evolution of lipid-binding properties. Specific CDR lipid-binding motifs are predicted to be encoded early in maturation processes for PGZL1 (*Zhang et al., 2019*), namely the CDR-H1 loop, and complemented by mutations at membrane-contacting FR-H3 and CDR-H3 loop residues incorporated later in maturation (*Figure 7—figure supplement 1C*). Across MPER bnAbs, membrane-binding molecular features are likely acquired at strategic timepoints throughout development to balance poly-specificity and evade autoimmune checkpoints (*Klein et al., 2013*). Experimental databases of germlines genes usage and of B-cell repertoires sequenced during immunization courses would be ideally paired with the simulations described here to investigate in vivo filtering of lineages with lipid-binding variants and assess possible rules of the autoimmune system (*Briney et al., 2019*).

Beyond gp41-targeting antibodies and lipid antibodies in autoinflammatory diseases (*Dema and Charles, 2016*; *Tao and Kriss, 1982*) or microbial infections (*Asherson and Cervera, 2003*; *Sène et al., 2008*), we suspect a broader scope of positive outcomes from antibody tolerance and interaction with lipid bilayers may occur in Nature, thus is currently underestimated and underutilized. Although host cross-specificity of 4E10 is well documented, the vastly reduced poly-reactivity of PGZL1 and 10E8 inspires optimism that maturation of phospholipid interactions and membrane tolerance may be incorporated into strategies for vaccine design and therapeutic antibodies for other membrane proteins with conserved buried epitopes (*Soto et al., 2016*). Currently, the breadth of analogous antibodies targeting membrane proteins via extensive or cooperative lipid interactions is poorly explored. The simulation procedures and principles described here are well positioned to investigate this outstanding question and could help define broader chemical rules for design of membrane-interfacing antibodies.

## Methods
### Atomistic simulations
The coordinates for Fab models were obtained from X-ray crystallography structures for each antibody (PDB id: 2FX9, 6O3D, 5T85 for 4E10, PGZL1, and 10E8 respectively). PDB 6SNE was used to generate LN01 + MPER TM systems. LN01 systems used PDB 6SND as a starting model. Any missing residues in atomic models were built with ModLoop. Original membrane docked orientations for each Fab in the bilayer were generated with Orientations of Proteins in Membranes (OPM) Positioning of Proteins in Membrane 2.0 (PPM) using per residue predicted transfer free energies from water to membrane environments (*Lomize et al., 2012*). For LN01 + MPER TM systems specifically, the PPM2.0 prediction used the entire protein complex for global docking optimization. Replicates with varied membrane docked starting orientations were generated by rotating the Fab along the first principal axis of the Fab +/- 15 degrees relative to the X axis.

Using the CHARMM-GUI webserver, the pre-positioned Fab was modified to include acetylated N-terminus (ACE) and methylamidated C-terminus (CT3) capping on each chain with appropriate disulfide bonds for a typical Fab structure. The Fab was then docked in an HIV-like lipid bilayer composed of 70% palmitoyl-oleoyl phosphatidylcholine (POPC), 25% cholesterol (CHOL), and 5% palmitoyl-oleoyl phosphatidic acid (POPA) bilayer and hydrated (minimum water height of 25 Å). Ions were added to neutralize and bring the system to a final concentrations of 0.15 mM KCl. Atomistic simulations were performed with Gromacs 2021 and CHARMM36 forcefield. Water molecules were described with the TIP3 model. All systems were minimized, with 5000 steps with steepest descent algorithm. A 2 fs time step was used along with the LINCS constraint algorithm during the equilibration stage. Electrostatics were treated with Particle Mesh Ewald, and the cutoff for both Coulomb and van der Waals interactions (Lennard Jones potential) was 1.2 nm. An 100 ps NVT equilibration phase applied a 4000 kJ/mol/nm$^3$ restraint on heavy atoms in protein and 1000 kJ/mol/nm$^3$ restraint on lipids with the velocity rescaling temperature coupling for protein, lipid, and solvent groups independently coupled to a 310 K bath with a temperature time constant of 0.1 ps$^{-1}$. A subsequence 15 ns NPT equilibration phase was performed with lipid restraints removed and 1000 kJ/mol/nm$^3$ restraints on the alpha carbons of protein. The Berendsen thermostat was used, with protein, lipid, and solvent independently coupled to a 310 K bath with temperature time constant 1.0 ps$^{-1}$. The Berendsen barostat was used with semi-isotropic coupling. Restraints were then removed for 0.5–1 µs production run with Noose-Hoover thermostat applied to protein, lipid, and solvent groups independently coupled to a 310 K bath with temperature time constant 1.0 ps$^{-1}$. The Parrrinello-Rahman barostat was used with semi-isotropic coupling and a 5.0 ps$^{-1}$ pressure time constant.

## Generating coarse grain models

X-ray structures were used as starting modes for each antibody (PDB id: 2FX9, 6O3D, 5T85 for 4E10, PGZL1, and 10E8, respectively). Coarse grain representations of Fab models were generated based on the Martini version 2.2 four to one mapping of heavy atoms to bead representation using the martinize script with an elastic network that defined harmonic bonds at default parameters. All coarse grain simulations were performed with Gromacs 2021 and CHARMM36 forcefield.

## Pre-embedded coarse grain simulations

Coarse grain Fab models were varied in their pre-embedded orientation by a rotation matrix of every 90 degrees in X axis, every 90 degrees in the Y axis, and every 1 nm along Z axis for 3 nm. A total of 40 Fab orientations that were physiologically plausible and did not include the constant region deeply embedded in the membrane were placed in a 15x15 × 20 nm box. After adding a Fab model to a box, Martini coarse grain lipids were added to the bottom half of the box at final percentage of 70% POPC, 25% cholesterol, and 5% POPA. Ions were added to neutralize the system to a final concentration of 0.15 mM NaCl and coarse grain water molecules were added to solvate the system with an adjusted van der Waals radius of 0.21 nm. A 30 fs time step was used along with the LINCS constraint algorithm. The system was minimized for 5000 with steepest descent algorithm. Minimization was followed by a short 30 ns membrane assembly step where the cutoff for both Coulomb and van der Waals interactions (Lennard Jones potential) was 1.1 nm. The velocity rescaling thermostat was used, with protein, lipid, and solvent independently coupled to a 310 K bath with 1.0 ps$^{-1}$ temperature time constant. The Berendsen barostat was used with isotropic coupling and a 6.0 ps$^{-1}$ pressure time constant. A 5 us production simulation was then performed with velocity rescaling thermostat applied to protein, lipid, and solvent groups independently coupled to a 310 K bath with temperature time constant 1.0 ps$^{-1}$. The Parrrinello-Rahman barostat was used with semi-isotropic coupling and 12.0 ps$^{-1}$ pressure time constant.

## Spontaneously associating coarse grain simulations

We used the same starting models and rotation matrix as in pre-embedded systems, but adjusted Z height to be 1–2 nm above a membrane plane. From these models, 18 starting Fab orientations were generated. A 70% POPC, 25% cholesterol, and 5% POPA coarse grain membrane was preassembled in a 30 ns simulation as describe in the pre-assembled pipeline. The assembled lipids and cholesterol molecules were extracted and placed into a 15x15 × 20 nm box along with a Fab molecule. This system was minimized in a vacuum using steepest descent algorithm, 30 fs time step, Berendsen

thermostat at 300 K with time constant 1 ps$^{-1}$, and Berendsen barostat with isotropic coupling and time constant 12 ps$^{-1}$. Ions were added to 0.15 Mm final concentration and the system was then solvated with water molecules using a van der Waals radius of 0.21 nm. A minimization step was repeated for the solvated system. The 100 ns equilibration phase was performed with a 10 fs step. The velocity rescaling thermostat was used, with protein, lipid, and solvent independently coupled to a 300 K bath and a temperature coupling time constant of 2.0 ps$^{-1}$. The Berendsen barostat was used with semi-isotropic coupling and a pressure coupling time constant of 12 ps$^{-1}$. A 14 μs production run was performed with 30 fs time step and velocity rescaling thermostat that coupled protein, lipid, and solvent independently to a 310 K bath with temperature coupling time constant of 1.0 ps$^{-1}$. The Berendsen barostat was used with semi-isotropic coupling and a pressure coupling time constant of 12 ps$^{-1}$.

## Headgroup binding site characterization

The X-ray structures of phosphate bound fab structures were used as references for alignments (PDB id: 4XCN, 6O3J, 5T80, 6SND for 4E10, PGZL1, 10E8, and LN01 respectively).

A metric to evaluate the accuracy to phosphate interaction in MD simulations and X-ray structures was calculated for every 0.5 ns of simulation time by superposing the interacting backbone residues of the MD simulation frame to interacting backbone residues from X-ray structure. For 4E10 and PGZL1, this superposition was done using CDRH1 residues 25–39. The RMSD was calculated between MD simulation's closest lipid headgroup phosphate coordinates and the phosphate in X-ray structure coordinates. For the 10E8 MD observed site, the FRL3 residues 83–87 (IMGT numbering) were aligned from every 0.5 ns of simulation time to the X-ray structure. After superposition, the root mean square distance (RMSD) was calculated between the closest lipid headgroup choline coordinates of the simulation frame and the phosphate coordinates in CDR-L1 loop of X-ray structure.

A bound state was then defined for each identified headgroup binding site. For 4E10 and PGZL1, a phosphate coordinate RMSD value and the distances of the interacting phosphate group to the loop backbone and side chain hydrogen atoms that were potential hydrogen bond donors were also measured. A phosphate group-hydrogen atom distance less than 5.25 Å was considered a potentially satisfied hydrogen bond interaction after accounting for thermal fluctuation and the mass averaged reference point for atoms of the phosphate group. For simulation frames where the phosphate coordinate RMSD was less than 2.0 Å (or 3.5 Å for the 10E8 MD predicted site) and at least two hydrogen bond interactions were considered potentially satisfied, that specific 0.5 ns interaction was deemed as a bound phosphate state.

For LN01 and LN01 + MPER TM X-ray site characterization, we aligned the MD frame CDR-L1 residues 23–35 to X-ray structures. A potential cation-pi cage interaction was defined as 2 or more distances between the closest lipid choline and center of aromatic rings for residues Tyr32, Tyr100G, Trp680, and Tyr681 being less than 5.5 Å. Polar interactions with a distance less than 5.25 Å between side chain hydroxyl groups of Thr30, Tyr32, Ser100F, and Tyr100G and closest lipid choline groups were also recorded. A hydrogen bond interaction was defined as a distance less than 5.25 Å between the Lys31 and the lipid phosphate. Headgroup occupancy was defined by satisfying one of the following conditions: a complete cation-pi cage formed around the choline headgroup, 2 or more satisfied interactions between lipid choline and nearby polar residues, or bivalent interaction including a hydrogen bond with lipid phosphate and one or more polar choline interaction or one or more cation-pi interactions. X-ray site RMSD was calculated with the MD lipid choline coordinates versus the X-ray structures lipid choline moiety.

The loading site interaction was also defined by aligning the MD frame CDR-L1 residues 23–35 to X-ray structure. A cation-pi cage was evaluated with the 2 or more distances less than 5.5 Å between lipid headgroup choline and residues Tyr49, Try52, Tyr100g, Trp100h and Tyr100i. Polar interactions between hydroxyl groups of Tyr49, Tyr52, Thr53, Thr100c, Ser100d, Tyr100g, and Tyr100i and lipid headgroup choline were defined with distances less than 5.25 Å. Lipid phosphate hydrogen bond interactions with Trp680, Lys683, and Ser100 were defined with a distances less than 5.25 Å. A loading-site-bound state was defined with similar logic to X-ray site occupancy and required satisfying one of the conditions listed above using the loading site specific residues described here. In LN01 systems, MPER-TM residues used in headgroup occupancy definitions were simply excluded from consideration. Loading site RMSD was calculated with the MD lipid

choline coordinates versus a reference position of the averaged coordinates for bound choline headgroups.

## Antibody fragment RMSF calculations

To calculate small scale changes in loop conformations, the MD simulation CDR-H1 residues were aligned from every 0.5 ns of MD simulated 4E10 and PGZL1 to relevant X-ray structures. FR-L3 residues from MD simulations to were aligned to 10E8 FRL3 X-ray structures, and CDR-L1 and CDR-H3 loops were aligned to LN01 X-ray structures by superposing the same residues described in headgroup RMSD calculations. A set of reference coordinates was calculated by averaging the loop or framework backbone atom coordinates across the aggregated of simulation time. We then calculated the RMSD of the MD frame backbone atoms to the average reference coordinates of backbone atoms after superposition. The RMSF values were subset into bound and unbound time based on headgroup occupancy across trajectories.

## Geometric descriptor calculations

### Angles

To calculate the angle to the membrane, two vectors along the pseudo-symmetry axes of the Fab and a normal vector to membrane plane were defined. The angle of approach vector is defined through two points: a point in the center of the tips of CDRL3 and CDRH3 loops, and a Fab center point between the proline residues (residue 40 with IMGT numbering) in heavy and light chains. The angle of rotation vector is defined with a point from the heavy chain hinge region through the light chain hinge region. The normal to the membrane plane is defined by fitting a plane to the phosphates in the top leaflet of the lipid bilayer and calculating a normal vector. Using these three vectors, the angle between the membrane plane and the antibody approach or rotation vector were calculated. All angles were transformed to be between –90 and 90 for angle of rotation and 0 and 90 for angle of approach.

### CDR loop depths

The depth of CDR loops was calculated by finding the Z height of the mass averaged CDR loop center point (for L1, L2, L3, H1, H2, H3) and measuring the distance to the average Z height of the phosphates in the top lipid bilayer. CDR loop residues were selected based on IMGT numbering definitions.

### Membrane embedded surface area

The membrane embedded surface area was calculated by measuring a fully solvated Fab solvent accessible surface area (SASA) and subtracting the membrane embedded Fab SASA every 0.5 ns of atomistic simulation time. SASA was calculated with VMD measure plugin which implements a point-based random sampling approach with 1.4 nm sphere.

### Interaction profiles

Interaction profiles were generated by defining the depth of each residue relative to the lipid bilayer over every 0.5 ns for atomistic simulations or every 30 ns for coarse grain simulations. The phosphate layer depth range was defined by calculating the average Z height of the top leaflet phosphates and expanding +/- 3 Å to account for headgroup size and thermal fluctuations in space. The glycerol layer was defined from the bottom of the phosphate layer – 4 Å based on glycerol molecule heights. The hydrocarbon layer was defined as up to 8 Å angstroms below the bottom of the glycerol layer. Because of the four to one mapping of atoms to beads in coarse grain representations, the layer definitions were expanded by 4 Å for coarse grain interaction profiling. Only the top leaflet layers were defined as a Fab was not observed to embedded deeper than below the top leaflet. Any non-lipid interacting residues were mapped to the solution layer of the system. The position of each residue in the Fab was mapped to a defined layer in the system based on residue Z height and time in each layer was aggregated over the total time of interest.

### Spontaneous membrane association

To define an antibody that associated with the membrane in coarse grain systems, the distance of each Fab reside to the phosphate plane in the top or bottom bilayer was measures. This allowed us

to account for a Fab that may diffuse across periodic boundaries in the defined system box. Tracking Fab residue positions every 30 ns, if a residue was within 4 Å of the phosphate layer, it was considered a Fab-membrane association.

## Structural bioinformatics

The PICES server was used to curate a list of 33,372 X-ray structure PDB entries with resolution below than 2.5 Å and less than 90% sequence similarity were downloaded into an in-house database. Method of Accelerated Search for Tertiary Ensemble Representatives (MASTER) was used to query protein structural fragments against the custom representative database to return hits with a maximum backbone RMSD of 1.8 Å. These backbone matches were then re-queried to find hits containing sulfate or phosphate anions within 1.5 Å RMSD from the initial query phosphate.

## Clustering and medoid definitions

Feature-based clustering was performed on the atomistic or coarse grain simulations. An 8-feature vector was defined for every 0.5 ns in atomistic simulation time that was composed of angle of approach, angle of rotation, and CDRHL1, CDRL2, CDRL3, CDRH1, CDRH2 and CDRH3 loop center of mass depths. In these clustering approaches, hierarchical agglomerative clustering with Ward's linkage method was used to minimize the variance between clusters defined. Euclidian distance thresholds were selected for each clustering experiment (antibody system and simulation scheme) to define clusters that optimize (macroscopic substates) or minimize (microscopic substates) divergence for average geometric descriptors across clusters in the context total clustering space.

To reduce the computational cost of clustering coarse grain simulation data, the feature vector was reduced to only include angle of rotation and angle of approach and sampled every 30 ns of simulation time. From here the same clustering methods as described before were applied and defined Euclidian distance cutoffs to generate three clusters from the sampled time for coarse grain systems. Applied K-medoids clustering with an n equal to 1 was used to identify medoids that are the best representative snapshots of clusters.

## Coarse grain to all-atom backmapping

Spontaneous insertion coarse grain clusters for 4E10, PGZL1, and 10E8 were defined using vectors describing each frame by angle of approach and angle of rotation. Hierarchical agglomerative clustering was performed with Ward's linkage method was used to minimize the variance between clusters. Euclidian distance thresholds were selected for so that three clusters were defined in for each antibody. Representative medoid frames were selected by calculating the frame that minimizes within-cluster sum of squares with Kmedoids algorithm from scikit-learn Clustering module.

To revert medoid frames from spontaneously associated coarse grain substates into atomistic representations, the original atomic Fab model (PDB ids: 2FX9, 6O3D, 5T85 for 4E10, PGZL1, and 10E8, respectively) was oriented relative to an assembled atomistic membrane (70% POPC, 25% Chol, 5% POPA) based on the corresponding coarse grain geometric representation by transforming the angle of approach and angle of rotation vectors. After docking the Fab, clashing lipid molecules were removed. Systems were solvated with TIP3 water molecules and neutralized with ions to a total concentrations of 0.15 mM NaCl. Minimization, equilibration, and production simulations were performed as previously described for atomistic simulations.

## Constant velocity pulling simulations

To generate starting conformation ensembles, atomic Fab models were docked into HIV-like lipid bilayers with the Orientations of Proteins in Membranes (OPM) Positioning of Proteins in Membrane 2.0 (PPM) server. 500 ns equilibration simulations were performed as described in Atomistic Simulations and frames were extracted every 50 ns. The same Fab models were used for 4E10, PGZl1, and 10E8 as in primary atomistic simulations. A PGZL1 intermediate model was constructed by building loops into PDB ID 6O41 with ModLoop. PGZL1 germline model was built by introducing mutations into the PGZL1 intermediate model in PyMOL. Both PGZL1 germline and PGZL1 intermediate models were simulated for 50ns in solution with the same protocol in Atomistic Simulations prior to docking into bilayers and 500 ns equilibrations.

For pulling trajectories, starting frames were simulated with 1000 kJ/mol/nm$^3$ restraint on lipids for 100 ns production runs. The Berendsen thermostat was applied to protein, lipid, and solvent groups independently coupled to a 310 K bath with temperature time constant 1.0 ps$^{-1}$. The Berendsen barostat was used with semi-isotropic coupling and a 5.0 ps$^{-1}$ pressure time constant. Center of mass pulling was turned on between protein and lipid groups. Umbrella sampling potential was applied between the two groups at a rate of 0.03 nm/ns with a force constant of 1000 kJ/mol/nm in the direction away from the bilayer. Statistical comparisons for comparing max forces across systems were using with Tukey's multiple comparison test and unpaired t-tests.

## Acknowledgements

The authors would like to acknowledge the High-Performance Computing Core at Scripps Research and the technical assistance of JC Ducom. Michael B Zwick and Daniel Leaman provided inspiration and helpful discussions. CAM was supported by the John and Susan Diekman Skaggs Graduate School Fellowship. This work was funded in part by Cooperative Agreement award UM1 AI144462 in partnership with the Division of AIDS, NIAID [IAW and ABW].

## Additional information

### Funding

| Funder | Grant reference number | Author |
|---|---|---|
| National Institute of Allergy and Infectious Diseases | UM1 AI144462 | Ian A Wilson<br>Andrew B Ward |
| Scripps Research Institute | John and Susan Diekman Skaggs Graduate School Fellowship | Colleen A Maillie |

The funders had no role in study design, data collection and interpretation, or the decision to submit the work for publication.

### Author contributions

Colleen A Maillie, Conceptualization, Data curation, Formal analysis, Writing – original draft, Writing – review and editing; Kiana Golden, Data curation; Ian A Wilson, Supervision, Writing – review and editing; Andrew B Ward, Resources, Supervision, Writing – review and editing; Marco Mravic, Conceptualization, Resources, Data curation, Formal analysis, Supervision, Writing – original draft, Writing – review and editing

### Author ORCIDs

Colleen A Maillie ⓘ https://orcid.org/0000-0001-7050-4464
Andrew B Ward ⓘ https://orcid.org/0000-0001-7153-3769
Marco Mravic ⓘ https://orcid.org/0000-0001-6294-1824

Reviewer #1 (Public review): https://doi.org/10.7554/eLife.90139.3.sa1
Reviewer #2 (Public review): https://doi.org/10.7554/eLife.90139.3.sa2
Author response https://doi.org/10.7554/eLife.90139.3.sa3

## Additional files

### Supplementary files
MDAR checklist

### Data availability
Representative simulation coordinate files and example equilibration and production files for simluation set up are uploaded to Zenodo. Simulation analysis scripts are available on GitHub (copy archived at *Maillie, 2025*).

The following dataset was generated:

| Author(s) | Year | Dataset title | Dataset URL | Database and Identifier |
|---|---|---|---|---|
| Maillie CA | 2023 | MPER bnAb Simluations | https://doi.org/10.5281/zenodo.13830877 | Zenodo, 10.5281/zenodo.13830877 |

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
