## [Editor Report · eLife Assessment]

This **valuable** study reports multi-scale molecular dynamics simulations to investigate a class of highly potent antibodies that simultaneously engage with the HIV-1 Envelope trimer and the viral membrane. The work provides insights into how broadly neutralizing antibodies associate with lipids proximal to membrane-associated epitopes to drive neutralization. After extensive revision, the level of evidence is considered **solid**, although a quantitative assessment of the underlying energetics remain difficult to obtain.

---

## [Referee Report · Reviewer #1 (Public review)]

Previous experimental studies demonstrated that membrane association drives avidity for several potent broadly HIV-neutralizing antibodies and its loss dramatically reduces neutralization. In this study, the authors present a tour de force analysis of molecular dynamics (MD) simulations that demonstrate how several HIV-neutralizing membrane-proximal external region (MPER)-targeting antibodies associate with a model lipid bilayer.

First, the authors compared how three MPER antibodies, 4E10, PGZL1, and 10E8, associated with model membranes, constructed with two lipid compositions similar to native viral membranes. They found that the related antibodies 4E10 and PGZL1 strongly associate with a phospholipid near heavy chain loop 1, consistent with prior crystallographic studies. They also discovered that a previously unappreciated framework region between loops 2-3 in the 4E10/PGZL1 heavy chain contributes to membrane association. Simulations of 10E8, an antibody from a different lineage, revealed several differences from published X-ray structures. Namely, a phosphatidylcholine binding site was offset and includes significant interaction with a nearby framework region. The revised manuscript demonstrates that these lipid interactions are robust to alterations in membrane composition and rigidity. However, it does not address the reverse-that phospholipids known experimentally not to associate with these antibodies (if any such lipids exist) also fail to interact in MD simulations.

Next, the authors simulate another MPER-targeting antibody, LN01, with a model HIV membrane either containing or missing an MPER antigen fragment within. Of note, LN01 inserts more deeply into the membrane when the MPER antigen is present, supporting an energy balance between the lowest energy conformations of LN01, MPER, and the complex. These simulations recapitulate lipid binding interactions solved in published crystallographic studies but also lead to the discovery of a novel lipid binding site the authors term the "Loading Site", which could guide future experiments with this antibody.

The authors next established course-grained (CG) MD simulations of the various antibodies with model membranes to study membrane embedding. These simulations facilitated greater sampling of different initial antibody geometries relative to membrane. These CG simulations , which cannot resolve atomistic interactions, are nonetheless compelling because negative controls (ab 13h11, BSA) that should not associate with membrane indeed sample significantly less membrane.

Distinct geometries derived from CG simulations were then used to initialize all-atom MD simulations to study insertion in finer detail (e.g., phospholipid association), which largely recapitulate their earlier results, albeit with more unbiased sampling. The multiscale model of an initial CG study with broad geometric sampling, followed by all-atom MD, provides a generalized framework for such simulations.

Finally, the authors construct velocity pulling simulations to estimate the energetics of antibody membrane embedding. Using the multiscale modelling workflow to achieve greater geometric sampling, they demonstrate that their model reliably predicts lower association energetics for known mutations in 4E10 that disrupt lipid binding. However, the model does have limitations: namely, its ability to predict more subtle changes along a lineage-intermediate mutations that reduce lipid binding are indistinguishable from mutations that completely ablate lipid association. Thus, while large/binary differences in lipid affinity might be predictable, the use of this method as a generative model are likely more limited.

The MD simulations conducted throughout are rigorous and the analysis are extensive, creative, and biologically inspired. Overall, these analyses provide an important mechanistic characterization of how broadly neutralizing antibodies associate with lipids proximal to membrane-associated epitopes to drive neutralization.

---

## [Referee Report · Reviewer #2 (Public review)]

In this study, Maillie et al. have carried out a set of multiscale molecular dynamics simulations to investigate the interactions between the viral membrane and four broadly neutralizing antibodies that target the membrane proximal exposed region (MPER) of the HIV-1 envelope trimer. The simulation recapitulated in several cases the binding sites of lipid head groups that were observed experimentally by X-ray crystallography, as well as some new binding sites. These binding sites were further validated using a structural bioinformatics approach. Finally, steered molecular dynamics was used to measure the binding strength between the membrane and variants of the 4E10 and PGZL1 antibodies.

The use of multiscale MD simulations allows for a detailed exploration of the system at different time and length scales. The combination of MD simulations and structural bioinformatics provides a comprehensive approach to validate the identified binding sites. Finally, the steered MD simulations offer quantitative insights into the binding strength between the membrane and bnAbs.

While the simulations and analyses provide qualitative insights into the binding interactions, they do not offer a quantitative assessment of energetics. The coarse-grained simulations exhibit artifacts and thus require careful analysis.

This study contributes to a deeper understanding of the molecular mechanisms underlying bnAb recognition of the HIV-1 envelope. The insights gained from this work could inform the design of more potent and broadly neutralizing antibodies.

---

## [Author Response]

The following is the authors’ response to the current reviews.

**Reviewer #1 (Public review):**
Previous experimental studies demonstrated that membrane association drives avidity for several potent broadly HIV-neutralizing antibodies and its loss dramatically reduces neutralization. In this study, the authors present a tour de force analysis of molecular dynamics (MD) simulations that demonstrate how several HIV-neutralizing membrane-proximal external region (MPER)-targeting antibodies associate with a model lipid bilayer.First, the authors compared how three MPER antibodies, 4E10, PGZL1, and 10E8, associated with model membranes, constructed with two lipid compositions similar to native viral membranes. They found that the related antibodies 4E10 and PGZL1 strongly associate with a phospholipid near heavy chain loop 1, consistent with prior crystallographic studies. They also discovered that a previously unappreciated framework region between loops 2-3 in the 4E10/PGZL1 heavy chain contributes to membrane association. Simulations of 10E8, an antibody from a different lineage, revealed several differences from published X-ray structures. Namely, a phosphatidylcholine binding site was offset and includes significant interaction with a nearby framework region. The revised manuscript demonstrates that these lipid interactions are robust to alterations in membrane composition and rigidity. However, it does not address the reverse-that phospholipids known experimentally not to associate with these antibodies (if any such lipids exist) also fail to interact in MD simulations.Next, the authors simulate another MPER-targeting antibody, LN01, with a model HIV membrane either containing or missing an MPER antigen fragment within. Of note, LN01 inserts more deeply into the membrane when the MPER antigen is present, supporting an energy balance between the lowest energy conformations of LN01, MPER, and the complex. These simulations recapitulate lipid binding interactions solved in published crystallographic studies but also lead to the discovery of a novel lipid binding site the authors term the "Loading Site", which could guide future experiments with this antibody.The authors next established course-grained (CG) MD simulations of the various antibodies with model membranes to study membrane embedding. These simulations facilitated greater sampling of different initial antibody geometries relative to membrane. These CG simulations , which cannot resolve atomistic interactions, are nonetheless compelling because negative controls (ab 13h11, BSA) that should not associate with membrane indeed sample significantly less membrane.Distinct geometries derived from CG simulations were then used to initialize all-atom MD simulations to study insertion in finer detail (e.g., phospholipid association), which largely recapitulate their earlier results, albeit with more unbiased sampling. The multiscale model of an initial CG study with broad geometric sampling, followed by all-atom MD, provides a generalized framework for such simulations.Finally, the authors construct velocity pulling simulations to estimate the energetics of antibody membrane embedding. Using the multiscale modelling workflow to achieve greater geometric sampling, they demonstrate that their model reliably predicts lower association energetics for known mutations in 4E10 that disrupt lipid binding. However, the model does have limitations: namely, its ability to predict more subtle changes along a lineage-intermediate mutations that reduce lipid binding are indistinguishable from mutations that completely ablate lipid association. Thus, while large/binary differences in lipid affinity might be predictable, the use of this method as a generative model are likely more limited.The MD simulations conducted throughout are rigorous and the analysis are extensive, creative, and biologically inspired. Overall, these analyses provide an important mechanistic characterization of how broadly neutralizing antibodies associate with lipids proximal to membrane-associated epitopes to drive neutralization.
**Reviewer #2 (Public review):**
In this study, Maillie et al. have carried out a set of multiscale molecular dynamics simulations to investigate the interactions between the viral membrane and four broadly neutralizing antibodies that target the membrane proximal exposed region (MPER) of the HIV-1 envelope trimer. The simulation recapitulated in several cases the binding sites of lipid head groups that were observed experimentally by X-ray crystallography, as well as some new binding sites. These binding sites were further validated using a structural bioinformatics approach. Finally, steered molecular dynamics was used to measure the binding strength between the membrane and variants of the 4E10 and PGZL1 antibodies.The use of multiscale MD simulations allows for a detailed exploration of the system at different time and length scales. The combination of MD simulations and structural bioinformatics provides a comprehensive approach to validate the identified binding sites. Finally, the steered MD simulations offer quantitative insights into the binding strength between the membrane and bnAbs.While the simulations and analyses provide qualitative insights into the binding interactions, they do not offer a quantitative assessment of energetics. The coarse-grained simulations exhibit artifacts and thus require careful analysis.This study contributes to a deeper understanding of the molecular mechanisms underlying bnAb recognition of the HIV-1 envelope. The insights gained from this work could inform the design of more potent and broadly neutralizing antibodies.
**Recommendations for the authors:**

**Reviewing Editor:**
We recommend the authors remove the figure and section related to bnAb LN01, perform additional analysis (e.g., further expanding on the differences in antibody binding in the presence or absence of antigen), and present this as a separate manuscript in a follow-up study.

We consider the analysis of a bnAb with a transmembrane antigen and of LN01 as essential to the manuscript and novel results. Study of LN01 provides many insights unique from the other MPER bnAbs in this study. We agree further characterization of LN01 and bnAbs with transmembrane antigen or full-length Env are intriguing and necessary to complete the full mechanistic understanding of lipid-associated antibodies. LN01 section in this paper is novel in the field and demonstrates the preliminary evidence motivating further work, which we agree are beyond the scope of this already long detailed study.

**Reviewer #1 (Recommendations for the authors):**
I appreciate the degree to which the authors responded to my previous points raised in the private review, including edits where I might have missed something in the manuscript or relevant literature. I imagine such a point-by-point response was quite onerous. Thank you also for balancing presentation/clarity with content/rigor considering the large information content of this manuscript; in silico results are inherently hard to present given the delicate balance between rigorous validation and novel information content. I apologize if I repeat points raised and addressed previously and commend the authors on their revised study, which is much improved in clarity; any additional revisions are of course entirely at your discretion."...now having more diversity in lipid headgroup chemistries" references the wrong figure-it should be: Figure 2-figure supplement 2A-C. The incorrect figure is also referenced again several sentences down: "...relevant CDR and framework surface loops..."

Thank you for pointing out this error. We have corrected figure references.

"One shared conformational difference observed for these bnAbs the higher cholesterol bilayers was slightly more extensive and broader interaction profiles as well as modestly deeper embedding of the relevant CDR and framework surfaces loops" please rephrase

Thank you for this suggestion. We rephrased this for improved clarity and flow.

"These results bolster the feasibility for using all-atom MD as an in silico platform to explore differential phospholipid affinity at these sites (i.e., specificity studies) and influence on antibody preferred conformation as membrane composition and lipid chemistry are systematically varied" Please tone down these speculations-you have demonstrated that simulations are robust to different headgroup chemistries but have not provided evidence for the exclusion of lipids that are known not to associate with these antibodies.

We rephrased this speculation to highlight the potential of this application. We also emphasize future studies that would be required to achieve this application in the following sentence.

“These results motivate use of all-atom MD as an in silico approach for exploring differential phospholipid affinity at these sites…”

Figure 2A: Specify which PDB entry corresponds to the displayed crystal structures in the main figure or caption.

We clarified these PDB entries in the figure caption.

Check reference formatting in supplemental figures when generating VOR.I am not sure how relevant this might be to the claims of Figure 2-figure supplement 3, but AlphaFold3-based phospholigand docking might provide an additional orthogonal approach if relevant ligand(s) are available for such analysis (particularly for the newly proposed 10E8 POPC complex).

Thank you for this suggestion. AI/ML based prediction methods like AF3 and RoseTTAFold All-Atom (RFAA) are interesting new methods that have come since our initial submission. We’ve decided these experiments are beyond the scope of this already long and detailed study. We have added a sentence suggesting use of these methods in future work.

"We next studied bnAb LN01 to interrogate differences"  this transition still reads a bit unclear. Why shift gears and change antibodies? Also, while you do go into its interactions both +/- antigen, there's no lead into the simulation initialization with and without antigen to guide the reader into the comparisons you will draw in the figure. Also, the order of information presentation is a bit strange, where the rationale for choosing a single monomeric helix is brought up in the middle of the paragraph instead of at the beginning of the section. In the next paragraph, it goes back to the initialization of the membrane composition again, which feels a bit disorganized-I do appreciate the unique challenge of having to weave through so much quality data! In fact, if you were to conduct simulations of membrane + antigen vs. membrane + LN01 vs. membrane + LN01 + antigen, I am tempted to say that this could be removed from this manuscript and flow better as a paper in and of itself.

We thank the reviewer for the suggestion to improve the writing style. We feel this section adds a lot of value to the manuscript, so we will keep it in the paper and improved the transition as well as rationale.

We selected to study the additional antibody LN01 and the monomeric MPER-TM antigen conformation because of the existing structural evidence available without additional creative model building. This rationale has been updated in the new text.

We changd the order of information as suggested, moving the rationale for antigen fragment earlier in the paragraph followed by the background of the lipids sites from the crystal that can lead into simulation set-up. We clarified the simulation initialization was similar for systems with and without antigen in the opening sentence of the paragraph

"previously observed snorkeling and hydration of TM Arg686"  Is this R696 (numbering could be different based on the particular Env)?

Thank you for noting this typo, we have corrected the numbering.

Potential font color issue with Figure 3-Figure supplement 1 B and part of A text-could be fixed in typesetting.The discussion reads very well. Is it possible to direct antibody maturation, even in an engineered context, towards membrane affinity without increasing immunogenic polyreactivity? This is mentioned very briefly and cited with ref 36, but I would be interested in the author's thoughts on this topic.

We thank the reviewer for the insightful idea to explore in future work. Our conclusion alludes to possibly artificially evolving membrane affinity studied by MD, as done in vitro by Nieva and co-workers. Because the hypothetical nature, we’ve chosen not to elaborate on those ideas from this manuscript.

**Reviewer #2 (Recommendations for the authors):**
To ensure reproducibility and facilitate further research, the authors should publicly deposit the code for running the MD simulations and analyses (e.g., on GitHub) along with the underlying data used in the study (e.g., on Zenodo.org).

We appreciate the consideration for open-source code and analysis. Representative code and simulation trajectories were uploaded to the following repositories:

https://github.com/cmaillie98/mper_bnAbs.git

https://zenodo.org/records/13830877

—-

The following is the authors’ response to the original reviews.

**Public Reviews:**

**Reviewer #1 (Public Review):**
Previous experimental studies demonstrated that membrane association drives avidity for several potent broadly HIV-neutralizing antibodies and its loss dramatically reduces neutralization. In this study, the authors present a tour de force analysis of molecular dynamics (MD) simulations that demonstrate how several HIV-neutralizing membrane-proximal external region (MPER)-targeting antibodies associate with a model lipid bilayer.First, the authors compared how three MPER antibodies, 4E10, PGZL1, and 10E8, associated with model membranes, constructed with a lipid composition similar to the native virion. They found that the related antibodies 4E10 and PGZL1 strongly associate with a phospholipid near heavy chain loop 1, consistent with prior crystallographic studies. They also discovered that a previously unappreciated framework region between loops 2-3 in the 4E10/PGZL1 heavy chain contributes to membrane association. Simulations of 10E8, an antibody from a different lineage, revealed several differences from published X-ray structures. Namely, a phosphatidylcholine binding site was offset and includes significant interaction with a nearby framework region.Next, the authors simulate another MPER-targeting antibody, LN01, with a model HIV membrane either containing or missing an MPER antigen fragment within. Of note, LN01 inserts more deeply into the membrane when the MPER antigen is present, supporting an energy balance between the lowest energy conformations of LN01, MPER, and the complex. Additional contacts and conformational restraints imposed by ectodomain regions of the envelope glycoprotein, however, remain unaddressed-the size of such simulations likely runs into technical limitations including sampling and compute time.The authors next established course-grained (CG) MD simulations of the various antibodies with model membranes to study membrane embedding. These simulations facilitated greater sampling of different initial antibody geometries relative to membrane. Distinct geometries derived from CG simulations were then used to initialize all-atom MD simulations to study insertion in finer detail (e.g., phospholipid association), which largely recapitulate their earlier results, albeit with more unbiased sampling. The multiscale model of an initial CG study with broad geometric sampling, followed by all-atom MD, provides a generalized framework for such simulations.Finally, the authors construct velocity pulling simulations to estimate the energetics of antibody membrane embedding. Using the multiscale modelling workflow to achieve greater geometric sampling, they demonstrate that their model reliably predicts lower association energetics for known mutations in 4E10 that disrupt lipid binding. However, the model does have limitations: namely, its ability to predict more subtle changes along a lineage-intermediate mutations that reduce lipid binding are indistinguishable from mutations that completely ablate lipid association. Thus, while large/binary differences in lipid affinity might be predictable, the use of this method as a generative model are likely more limited.The MD simulations conducted throughout are rigorous and the analysis are extensive. However, given the large amount of data presented within the manuscript, the text would benefit from clearer subsections that delineate discrete mechanistic discoveries, particularly for experimentalists interested in antibody discovery and design. One area the paper does not address involves the polyreactivity associated with membrane binding antibodies-MD simulations and/or pulling velocity experiments with model membranes of different compositions, with and without model antigens, would be needed. Finally, given the challenges in initializing these simulations and their limitations, the text regarding their generalized use for discovery, rather than mechanism, could be toned down.Overall, these analyses provide an important mechanistic characterization of how broadly neutralizing antibodies associate with lipids proximal to membrane-associated epitopes to drive neutralization.
**Reviewer #2 (Public Review):**
In this study, Maillie et al. have carried out a set of multiscale molecular dynamics simulations to investigate the interactions between the viral membrane and four broadly neutralizing antibodies that target the membrane proximal exposed region (MPER) of the HIV-1 envelope trimer. The simulation recapitulated in several cases the binding sites of lipid head groups that were observed experimentally by X-ray crystallography, as well as some new binding sites. These binding sites were further validated using a structural bioinformatics approach. Finally, steered molecular dynamics was used to measure the binding strength between the membrane and variants of the 4E10 and PGZL1 antibodies.The conclusions from the paper are mostly well supported by the simulations, however, they remain very descriptive and the key findings should be better described and validated. In particular:It has been shown that the lipid composition of HIV membrane is rich in cholesterol [1], which accounts for almost 50% molar ratio. The authors use a very different composition and should therefore provide a reference. It has been shown for 4E10 that the change in lipid composition affects dynamics of the binding. The robustness of the results to changes of the lipid composition should also be reported.The real advantage of the multiscale approach (coarse grained (CG) simulation followed by a back-mapped all atom simulation) remains unclear. In most cases, the binding mode in the CG simulations seem to be an artifact.The results reported in this study should be better compared to available experimental data. For example how does the approach angle compare to cryo-EM structure of the bnAbs engaging with the MPER region, e.g. [2-3]? How do these results from this study compare to previous molecular dynamics studies, e.g.[4-5]?References(1) Brügger, Britta, et al. "The HIV lipidome: a raft with an unusual composition." Proceedings of the National Academy of Sciences 103.8 (2006): 2641-2646.(2) Rantalainen, Kimmo, et al. "HIV-1 envelope and MPER antibody structures in lipid assemblies." Cell Reports 31.4 (2020).(3) Yang, Shuang, et al. "Dynamic HIV-1 spike motion creates vulnerability for its membrane-bound tripod to antibody attack." Nature Communications 13.1 (2022): 6393.(4) Carravilla, Pablo, et al. "The bilayer collective properties govern the interaction of an HIV-1 antibody with the viral membrane." Biophysical Journal 118.1 (2020): 44-56.(5) Pinto, Dora, et al. "Structural basis for broad HIV-1 neutralization by the MPER-specific human broadly neutralizing antibody LN01." Cell host & microbe 26.5 (2019): 623-637.

Considering reviewer suggestions, we slightly reorganized the results section into specific sub-sections with headings and changed the order in which key results were presented to allow the subsequent analysis more accessible for readers. Supplemental materials were redistributed into eLife format, having each supplemental item grouped to a corresponding main figure. Many slightly detail modifications were made to figures (mostly supplemental items) without changing their character, such as clearer axes labels or revised annotations within panels.

The major additions within the results sections based on the reviews were:

(1) An expanded the comparison between our simulation analyses to previous simulations and to existing cryo-EM structural evidence for MPER antibodies’ membrane orientation the context of full-length antigen, resulting in new supplemental figure panels.

(2) New atomistic simulations of 10E8, PGZL1, and 4E10 evaluating the phospholipid binding predictions in a different lipid composition more closely modeling HIV membranes.

Minor edits to the analyses and interpretations include:

(1) Outlining the geometric components contributing to variance in substates after clustering the atomistic 10E8, 4E10, and PGZL1 simulations.

(2) Better defining the variance and durability of membrane interactions within and across systems in the coarse grain methods section.

(3) Removed interpretations in the original results sections regarding polyreactivity and energetics for MPER bnAbs that were not explicitly supported by data.

(4) More context of the prevenance of bnAb loop geometries in structural informatics section

(5) Rationale for the choice of the continuous helix MPER-TM conformation in LN01-antigen conformations, and citations to previous gp41 TM simulations.

(6) Removed language on the novelty of the coarse grain and steered pulling simulations as newly developed approaches; tempering the potential discriminating power and applications of those approaches, in light of their limitations.

The discussion was revised to provide more novel context of the results within the field, including discussing direct relevance of the simulation methods for evaluating immune tolerance mechanisms and into antibody engineering. We have shared custom scripts used for molecular dynamics analysis on github (https://github.com/cmaillie98/mper_bnAbs.git) and uploaded trajectories to a public repository hosted on Zenodo (https://zenodo.org/records/13830877).

**Recommendations for the authors:**
Below, I provide an extensive list of minor edits associated with the text and figures for the authors to consider. I provide these with the hope of increasing the accessibility of the manuscript to broader audiences but leave changes to the discretion of the authors.Text/clarityFigure 1 main textThe main text discussing Figure 1 is disorganized, making the analysis difficult to follow. I would suggest the following: moving the sentence, "4E10 and PG2L1 are structurally homologous" immediately after the paragraph discussing the simulation initiation. Then, add a sentence that directly compares their experimental affinity, neutralization, and polyreactivity of 4E10 and PG2L1 (later, an unintroduced idea pops up, "These patterns may in part explain 4E10's greater polyreactivity"). Next, lead into the discussion of the MD simulation data with something to the effect of: "Given these similarities, we first compared mechanisms of membrane insertion between 4E10 and PG2L1 to bolster confidence in our predictions". Later, the sentence "Across 4E10 and PGZL1 simulations, the bound lipid phosphates"

We thank the reviewer for the suggestion and we have restructured the beginning of the results to implement this style: to first introduce then discuss the comparative PGZL1 & 4E10 results, i.e. Figure 1 plus associated supplements.

In the background and the introduction text leading up to Figure 1, CDR-H3 is discussed at length, however, the first figure focuses almost entirely on how CDR-H1 coordinates a lipid phosphate headgroup. Are there experimental mutations in this loop that do not affect affinity (e.g., to a soluble gp41 peptide), but do affect neutralization (like the WAWA mutation for CDR-H3, discussed later)?

We have altered the Introduction (para 2) and Results (4E10/PGZL1 sub-section) to give more balanced discussion of CDRs H1 & H3. That includes referencing experimental data addressing the reviewer’s question; a PGZL1 clone H4K3 where mutations to CDRH1 were introduced and shown have minimal impact on affinity to MPER peptide via ELISA and BLI, but those mutant bnAbs had significantly reduced neutralization efficacy (PMC6879610).

The sentence "These phospholipid binding events were highly stable, typically persisting for hundreds of nanoseconds" should be moved down to immediately precede, "[However], in a PGZL1 simulation, we observed a". This would be a good place for a paragraph break following, "Thus, these bnABs constitutively", since this block of text is very long.Similarly, the sentence and parts of the section, "Likewise, the interactions coordinating the lipid phosphate oxygens at CDR-H1" more appropriately belongs immediately before or after the sentence, "Our simulations uncover the CDR-lipid interactions that are the most feasible".

Thank you for the detailed guidance in reorganizing the Figure 1 results. We followed the advice to directly compare 4E10 and PGZL1 results separately from 10E8, moving those sections of text appropriately. New paragraph breaks were added to improve accessibility and flow of concepts throughout the Results.

In the sentence, "our simulations uncover CDR-lipid interactions that are the most feasible and biologically relevant in the context of a full [HIV] lipid bilayer... validation to which of the many possible ions" à have you confidently determined lipid binding and positioning outside of the site validated in figure 1? Which site(s) are these referencing? The next two sentences then introduce two new ideas on the loop backbone stability then lead into lipid exchange, which is a bit jarring.

We have adjusted the language concerning the putative ions/lipids electron density across the many PGZL1 and 4E10 crystal structures, and additionally make the explicit point that we confidently determined the lack of lipid binding outside of the site focused on in Figure 1.

“… both bnAbs showed strong hotspots for a lipid phosphate bound within the CDR-H1 loops, with minimal phospholipid or cholesterol ordering around the proteins elsewhere. The simulated lipid phosphates bound within CDR-H1 have exceptional overlap with electron densities and atomic details of modelled headgroups from respective lipid-soaked co-crystal structures…”

Figure 2 main text"We similarly investigated bnAb 10E8" - Please make this a separate subheader, the block text is very long up to this point.

Thank you for the suggestion. We introduced a sub-header to separate work on 10E8 all-atom simulations.

"we observed a POPC complexed with... modelled as headgroup phosphoglycerol anions..." - please cite the references within the text.

Thank you for pointing out this missing reference, we added the appropriate reference.

"One striking and novel observation" - please remove the phrase "striking" throughout, for following best practices in scientific writing (PMC10212555)-this is generally well-done throughout.

We removed “striking” from our text per your suggestion.

"This CDR-L1 site highlights... (>500 fold) across HIV strains" - How much do R29 and Y32 also contribute to antigen binding and the conformation of this loop? These mutants also decreased Kd by approximately 20X, and based on the co-crystal structure with the TM antigen (PDB: 4XCC), seem to play a more direct role in antigen contact. Additionally, these residues should be highlighted on a figure, otherwise it's difficult to understand why they are important for membrane association.

We thank the reviewer for deep engagement to these supporting experimental details. The R29A+Y32A 10E8 mutant referenced in the text showed only 4-fold Kd increase, a modest change for an SPR binding experiment. Whereas R29E+Y32E 10E8 mutant resulted in 40x Kd increase, the “20x” the reviewer refers to. Both 10E8 mutants showed similar drastically reduced breadth and potency of over 2 orders of magnitude on average.

These mutated CDR-L1 residues are not directly involved in antigen contact and adopt the same loop helix conformation when antigen is bound. A minor impact on antigen binding affinity could be due altering pre-organization of CDR loops upon losing interactions from the Tyr & Arg sidechains - particularly Tyr31 in contact with CDR-H3.

As per the suggestion, clearer annotated figure panel denoting these sidechains has been added to Figure 2-Figure Supplement 1 for 10E8 analysis.

"Structural searches querying... identified between 10^5 and 2*10^6..." - why is this value represented as such a large range? Does this depend on the parameters used for analysis? Please clarify.Additionally, how prevalent are any random loop conformations compared to the ones you searched? It's otherwise difficult to attribute number of occurrences within the 2 A cutoff to biological significance, as this number is not put in context.

We appreciate the reviewers comment to contextualize the range and relative frequency of the bnAb loop conformations. RMSD and length of loop are the key parameters, which can be controlled by searching reference loops of similar length. The main point of the backbone-level searching is simply to imply the bnAb loops are not particularly rare when comparing loops of similar length.

We did as was suggested and added comparison to random loops of the same length to the main text, including a new Supplementary Table 4.

“…identified between 105 to 2∙106 geometrically similar sub-segments within natural proteins (<2 Å RMSD)40, reflecting they are relatively prevalent (not rare) in the protein universe, comparing well with frequency of other surface loops of similar length in antibodies (Supplementary Table 3).”

"We next examined the geometries" could start after its own new subheading. Moreover, while there's an emphasis on tilt for neutralization, there is not a figure clearly modelling the proposed Env tilt compared to the relatively planar bilayer. It would be helpful to have an additional panel somewhere that shows the orientation of the antibody (e.g., a representative pose) in the simulations relative to an appropriately curved membrane, Env, the binding conformation of the antibody to Env, and apo Env, given the tilting observed in PMID: 32348769 and theorized in PMC5338832. What additional conformational changes or tilting need to occur between the antibodies and Env to accomplish binding to their respective epitopes?

Thank you for outlining an interesting element to consider in our analysis of a multi-step binding mechanism for MPER antibodies. We added additional figure panels in the supplement to outline the similarities and differences between our simulations and Fabs with the inferred membranes in cryo-EM experiments of full-length HIV Env. The simulated Fabs’ angles are very similar with only minor tilting to match the cryo-EM antibody-membrane geometries.

We added Figure 1-figure supplement 1A & Figure 2-figure supplement 2A, and alter to text to reflect this:

“The primary difference is Env-bound Fabs in cryo-EM adopt slightly more shallow approach angles (~15_°_) relative to the bilayer normal. The simulated bnAbs in isolation prefer orientations slightly more upright, but presenting CDRs at approximately the same depth and orientation. Thus, these bnAbs appear pre-disposed in their membrane surface conformations, needing only a minor tilt to form the membrane-antibody-antigen neutralization complex.”

Env tilt dynamics and membrane curvature of natural virions may reconcile some of these differences. Recent in situ tomography of Full-length Env in pseudo-virions corroborates our approximation of flat bilayers over the short length scales around Env.

The sentence "we next examined the geometries" mentions "potential energy cost, if any, for reorienting...". However, there's no further discussions of geometry or energy cost within this section. Please rephrase, or move this figure to main and increase discussion associated with the various conformational ensembles, their geometry, and their phospholipid association.

As the reviewer highlights, the unbiased simulations and our analysis do not explicitly evaluate energetics. We removed this phrase, and now only allude to the minimal energy barrier between the similar geometric conformations, relative to the tilting & access requirements for antigen binding mechanism.

“The apparent barrier for re-orientation is likely much less energetically constraining than shielding glycans and accessibility of MPER”

".. describing the spectrum of surface-bound conformations" cites the wrong figure.

Thank you for noticing this error; we correct the figure reference to (Figure 2-figure supplement 4).

Please comment on the significance of how global clustering (Fig. S5A-C) was similar for 4E10 and PGZL1, but different for 10E8 (e.g., blue, orange, and yellow clusters for 4E10 and PHZL1 versus cyan, red, and green clusters for 10E8). As the cyan cluster seems to be much closer in Euclidian space to the 4E10/PGZL1 clusters, it might warrant additional analysis. What do these clusters represent in terms of structure/conformation? How do these clusters differ in membrane insertion as in (A)?

We are grateful you identify analysis in the geometric clustering section that may be of interest to other readers. We have added additional supplementary table (Table 2) to detail the CDR loop membrane insertion and global Fab angles which describe each cluster, to demonstrate their similarities and differences. We also better describe how global clustering was similar for 4E10 and PGZL1, but different for 10E8 in the relevant results section

The cyan cluster is not close in structure to 4E10/PGZL1 clusters. We note the original figure panel had an error. The updated Figure 2-supplement 4B shows the correct Euclidian distance hierarchy with an early split between 4e10/pgzl1 and 10e8 clusters.

Figure 3 main textThe start of this section, "We next studied bnAb LN01...", is a good place for a new subheader.

We have added an additional subheader here: Antigen influence on membrane bound conformations and lipid binding sites for LN01

There should be a sentence in the main text defining the replicate setup and production MD run time. Is the apo and complex based on a published structure? How do you embed the MPER? Is the apo structure docked to membrane like in 4E10? The MD setup could also be better delineated within the methods.

The first two paragraphs in this section have been updated to clarify the relevant simulations configuration and Fab membrane docking prediction details.

The procedure was the same for predicting an initial membrane insertion, albeit now we use the LN01-TM complex and the calculation will account for the membrane burial of the the TM domain and MPER fragment. As mentioned, LN01 is predicted as inserted with CDR loops insert similarly with or without the TM-MPER fragment. The geometry differs from PGZL1/4E10 and 10E8, denoted by the text.

Please comment on the oligomerization state of the antigen used in the MD simulation: how does the simulation differ from a crossed MPER as observed in an MPER antibody-bound Env cryo-EM structure (PMID: 32348769), a three-helix bundle (PMC7210310), or single transmembrane helix (PMC6121722)? How does the model MPER monomer embed in the membrane compared to simulations with a trimeric MPER (PMC6035291, PMID: 33882664)-namely, key arginine residues such as R696?

We thank the reviewer for pointing out critical underlying rationale for modeling this TM-MPER-LN01 complex which we have corrected in the revised draft. The range of potential conformations and display of MPER based on TM domain organization could easily be its own paper – we in fact have a manuscript in preparation on the topic.

The updated text expands the rationale for choosing the monomeric uninterrupted helix form of the MPER-TM model antigen (para 1 of LN01 section). The alternative conformations we did not to explore are called out, with references provided by the reviewer.

The discussion qualified that the MPER presentation is likely oversimplified here, noting MPER display in the full-length Env trimer will vary in different conformational states or membrane environments. However, the only cryo-EM structures of full-length ENV with TM domains resolved have this continuous helix MPER-TM conformation – seen both within crossing TM dimers or dissociated TM monomers.

Are there additional analyses that can validate the dynamics of the MPER monomer in the membrane and relative to LN01? Such as key contacts you would expect to maintain over the duration of the MD simulation?

We also increased description of this TM domain’s behavior, dynamics (tilt, orientation, Arg696 snorkeling, and complex w LN01) to provide a clearer picture of the simulation results – which aligns with past MD of the gp41 TM domain as a monomer (para 2 of LN01 section). As well, we noted key LN01-MPER contacts that were maintained.

How does the model MPER modulate membrane properties like lipid density and lipid proximities near LN01?

We checked and didn’t notice differences for the types of lipids (chol, etc) proximal to the MPER-TM or the CDR loops versus the bulk lipid bilayer distributions. Due to the already long & detailed nature of this manuscript, we elect not to include discussion on this topic.

Supplemental figure 1H-I would be better positioned as a figure 3-associated supplemental figure.

We rearranged to follow the eLife format and have paired supplemental panels with their most relevant main figures.

Figure 3F/H reference a "loading site" but this site is defined much later in the text, which was confusing.

Thank you for pointing out this source of confusion, we rearranged our discussion to reflect the order in which we present data in figures.

What evidence suggests that lipids "quickly exchange from the Loading site into the X-ray site by diffusion"? I do not gather this from Figure S1H/I.

We have rearranged the loading side and x-ray site RMSD maps in Figure 3-Figure supplement 1 to better illustrate how a lipid exchanges between these sites.

Figure 4 main textThe authors assert that in the CG simulations, restraints, "[maintain] Fab tertiary and quaternary structure". However, backbone RMSD does not directly assert this claim-an additional analysis of the key interfacial residues between chains, or geometric analysis between the chains, would better support this claim.

Thank you for pointing this point. We rephrased to add that the major sidechain contacts between heavy and light chain persist, in addition to backbone RMSD, to describe how these Fabs maintain the fold stably in CG representation.

In several cases, CG models sample and then dissociate from the membrane. In the text, the authors mention, "course-grained models can distinguishing unfavorable and favorable membrane-bound conformations". Is there a particular orientation that causes/favors membrane association and dissociation? This analysis could look at conformations immediately preceding association and dissociation to give clues as to what orientation(s) favor each state.

Thank you for suggesting this interesting analysis. Clustering analysis of associated states are presented in Figure 5, Figure 5-Figure Supplement 1, and Figure 6, which show all CDR and framework loop directed insertion. This feature is currently described in the main text.

We did not find strong correlation of specific orientations as “pre-dissociation” states or ineffective non-inserting “scanning” events. We revised the key sentence to reflect the major take away – that non-CDR alternative conformations did not insert and most of those having CDRs inserted in a different manner than all-atom simulations also were prone to dissociate:

“Given that non-CDR directed and alternative CDR-embedded orientations readily dissociate, we conclude that course-grained models can distinguish unfavorable and favorable membrane-bound conformations to an extent that provides utility for characterizing antibody-bilayer interaction mechanisms.”

Figure 6 main text"For 4E10, trajectories initiated from all three geometries..." only two geometries are shown for each antibody. Please include all three on the plot.

The plots include markers for all three geometries for 4E10, highlighted in stars or with letters on the density plots of angles sampled (Figure 6B,C)

"Aligning a full-length IgG... unlikely that two Fabs simultaneously..." Are there theoretical conformations in which two Fabs could simultaneously associate with membrane? If this was physiological or could be designed rationally, could an antibody benefit further from avidity?

Our modeling suggests the theoretical conformations having two Fabs on the membrane are infeasible. It’s even less likely multiple Env antigens could be engaged by one IgG. We have revised the text to express this more clearly.

Figure 7 main text"An intermediate... showed a modest reduction in affinity..." what affinity does PGZL1 have for this antigen?

The preceding sentence for this information: “Mature PGZL1 has relatively high affinity to the MPER epitope peptide (Kd = 10 nM) and demonstrates great breadth and potency, neutralizing 84% of a 130 strain panel “

FiguresFigure 1It would be helpful to have an additional panel at the top of this figure further zoomed out showing the orientation of the antibody (e.g., a representative pose) in the simulations relative to an appropriately curved membrane, Env, the binding conformation of the antibody to Env, and apo Env, given the tilting observed in PMID: 32348769 and theorized in PMC5338832. What additional conformational changes or tilting need to occur between the antibodies and Env to accomplish binding to their respective epitopes?

Thank you for the suggestion to include this analysis. We have added to the text reflecting this information, as well as making new supplemental panels for 4E10 and 10E8 that we compare simulated 4E10 and 10E8 Fab conformations to cryoEM density maps with Fabs bound to full-length HIV Env. Figure 1-figure supplement 1A & Figure 2-figure supplement 2A

In Figure 1, space permitting, it would be helpful to annotate the distances between the phosphates and side chains (similarly, for Figure S1A).

To avoid the overloading the Main figure panels with text, those relevant distances are listed in the methods sections. Those distances are used to define the “bound” lipid phosphate state. Generally, we note the interactions are within hydrogen bonding distance.

Annotating "Replicate 1" and "Replicate 2" on the left side of Figure 1C/D would make this figure immediately intuitive.

We have added these labels.

Figure caption 1C: Please clarify the threshold/definition of a contact used to binarize "bound" versus "unbound" (for example, "mean distance cutoff of 2A between the phosphate oxygen and the COM of CDR-H1") [on further reading of the methods section, this criterion is quite involved and might benefit from: a sentence that includes "see methods"]. Additionally, C could use a sentence explaining the bar such as in E, "Phosphate binding is mapped to above each MD trajectory" Please define FR-H3 in the figure caption for E/F.

We have added these details to the figure caption.

Because Figure 1 is aggregated simulation time, it would be helpful to also represent the data as individual replicates or incorporate this information to calculate standard deviations/statistics (e.g., 1 microsecond max using the replicates to compute a standard deviation).

We believe the current quantification & display of data via sharing all trajectories is sufficient to convey the major point for how often each CDR-phosholipid binding site it occupied. Further tracking and statistics of inter-atomic distances will likely be too tedious & add minimal value. There is some dynamics of the phosphate oxygens between the polar within the CDR site but our “bound” state definitions sufficiently describe the key participating interactions are made.

Figure 2For A, it would be helpful to annotate the yellow and blue mesh on the figure itself.

We have defined the orange phosphate and blue choline densities.

Also, where are R29 and Y32 relative to this site? In the X-ray panels, Y38 is not shown, and the box delineating the zoom-in is almost imperceptible.

Thank you for this suggestion to include those amino acids which are referenced in the text as critical sites where mutation impacts function. To clarify, Y32 is the pdb numbering for residue Y38 in IMGT numbering. We have added a panel to Figure 2-Figure Supplement 1 having a cartoon graphic of 10E8 loop groove with sidechains & annotating R29 and Y38, staying consistent with out use of IMGT numbering in the manuscript.

Figure 3It might read clearer to have "LN01+MPER-TM" and "LN01-Apo" in the middle of A/B and C/D, respectively, and a dotted line delineating the left and right side of the figure panels.

We have added these details to the figure for clarity for readers.

It would be helpful to show some critical interactions that are discussed in the text, such as the salt bridge with K31, by labeling these on the figure (e.g., in E-H).

We drafted figure panels with dashed lines to indicate those key interactions. However, they became almost imperceptible and overloaded with annotations that distracted from the overall details. For K31, the interaction occurs in LN01 crystal structures readers can refer to.

Why are axes cut off for J?

We corrected this.

Please re-define K/L plots as in Figure 1, and explain abbreviations.

We updated the figure caption to reflect these changes.

Figure 4The caption for panel A states that the Fab begins in solvent 1-2 nm above the bilayer, but the main text states 0.5-2 nm.

We have reconciled this difference and listed the correct distances: 0.5-2nm.

Please label the y-axis as "Replicate" for relevant figure panels so that they are more immediately interpretable.

This label has been added.

A legend with "membrane-associated" and "non-associated" within the figure would be helpful. Additionally, the average percent membrane associated, with a standard deviation, should be shown (Similar to 1C, albeit with the statistics).

This legend has been added. We also added the additional statistical metrics requested to strengthen our analysis.

The text references "10, 14, and 12 extended insertion events" for the three antibody-based simulations. How do you define "extended insertion events"? Would breaking this into average insertion time and standard deviation better highlight the association differences between MPER antibodies and controls, in addition to the variability due to difference random initialization?

We thank the reviewer for the insightful suggestion on how to better organize quantitative analysis to support the method. Supplemental Table 3 includes these numbers.

Figure 5The analysis in Fig. S6C could be included here as a main figure.

The drafted revised figure adding S6C to Figure 5 made for too much information. Likewise, putting this panel S6C separated it from the parent clustering data of S6B, so we decided to keep these figures separated. The S6 figure is now Figure 5-figure supplement 1.

Figure 6Please annotate membrane insertion on E as %.

These are phosphate binding RMSD/occupancy vs time. The panels are now too small to annotate by %. The qualitative presentation is sufficient at this stage. The quantitative % are listed in-line within text when relevant to support assertions made.

Please use the figure caption to explain why certain clusters (e.g., 10E8 cluster A, artifact, Fig. S6E) are not included in panel E.

We have added this information in the figure caption.

Figure 7Please show all points on the box and whisker plots (panels E and F), and perform appropriate statistical tests to see if means are significantly different (these are mentioned in the text, but should be annotated on the graph and mentioned within the figure caption).

We have changed these plots to show all data points along with relevant statistical comparisons. The figure captions describe unpaired t-test statistical tests used.

Figure S1G, H, and I do not belong here-they should be moved to accompany their relevant text section, which associates with Figure 3. It would be helpful to associate this with Figure 3 in the eLife format, "Figure 3-Supplemental Figure 1" or its equivalent.It's very difficult to distinguish the green and blue circles on panel G.

We darkened the shading and added outline for better visualization

Subfigure I is missing a caption, could be included with H: "(H,I) Additional replicates for LN01+TM (H) and LN01 (I)".

We corrected this as suggested.

Why is H only 3 simulations and not 4? Does it not have a lipid in the x-ray site? Also, the caption states "(top, green)" and "(bottom, cyan)", but the green vs. cyan figures are organized on the left and right. Additional labels within the figure would help make this more intuitive.If the point of H and I is to illustrate that POPC exchanges between the X-ray and loading sites, this is unclear from the figure. Consider clarifying these figures.

Thank you for describing the confusion in this figure, we have added labels to clarify.

Figure S2 (panels split between revised Figure 4 associated figure supplements)The LN01 figures should likely follow later so that they can associate with Figure 3, despite being a similar analysis.

We corrected supplements to eLife format so supplements are associated with relevant main figures.

Figure S3 (panels split between revised Figure 1 & 2 associated figure supplements)As hydrophobicity is discussed as a driving factor for residue insertion, it would be helpful to have a rolling hydrophobicity chart underneath each plot to make this claim obvious.

We prefer the current format, due to the worry of having too much information in these already data-rich panels. As well, residues are not apolar but are deeply inserted.

Figure S4 (panels split between revised Figure 1 & 2 associated figure supplements)It would be helpful to label the relevant loops on these figures.

We have labeled loops for clarity.

Do any of these loops have minor contacts with Env in the structure?

The 4E10 and PGZL1 CDRH-1 loop does not directly contact bound MPER peptides bound in crystal structures.

FRL-3 and CDR-H1 in 10E8 do not contact the MPER peptide antigen component based on x-ray crystal structures.

Do motif contacts with lipid involve minor contacts with additional loops other than those displayed in this figure?

The phosphate-loop interactions in motifs used as query bait here are mediated solely by the backbone and side chain interactions of the loops displayed. We visually inspected most matches and did not see any “consensus” additional peripheral interactions common across each potential instance in the unrelated proteins. The supplied Supplemental Table 2 contains the information if a reader wanted to conduct a detailed search.

Why is there such a difference between the loop conformation adopted in the X-ray structure and that in the MD simulation, and why does this lead to the large observed differences in ligand-binding structure matches?

We thank the reviewer for carefully noting our error in labeling of CDR loop and framework region input queries. We revised the labeling to clarify the issue.

The is minimal structural difference between the loops in x-ray and MD.

Figure S5 (Figure 2-Figure supplement 4)This figure is not colorblind friendly-it would be helpful to change to such a pallet as the data are interesting, but uninterpretable to some.

We have left this figure the same.

"Susbstates" - "Substates"

Corrected, thank you.

Panel B is uninterpretable-please break the axis so that the Euclidian distances can be represented accurately but the histograms can be interpreted.

We have adjusted axis for this plot to better illustrate the cluster thresholds.

The clusters in D-H should be analyzed in greater depth. What is the structural relevance of these clusters other than differences in phospholipid occupancy in (I)? Snapshots of representative poses for each cluster could help clarify these differences.

We have adjusted the text to describe the geometric differences in each of those clusters that result in the different exceptionally lower propensities for forming the key phospholipid interaction.

The figure caption should make it clear that 3 μS of aggregate simulation time is being used here instead of 4 μS to start with unique tilt initializations. E.g., "unique starting membrane-bound conformations (0 degrees, -15 degrees, 15 degrees initialization relative to the docked pose)". Further, why was the particular 0-degree replicate chosen while the other was thrown out? Or was this information averaged? Why is the full 4 μS then used for D-I?

We thank the reviewer for noting these details. We didn’t want to bias the differential between 10E8 and 4E10/PGZL1 by including the replicate simulations. The analysis was mainly intended to achieve more coarse resolution distinction between 10E8 and the similar PGZL1/4E10.

In the subsequent clustering of individual bnAb simulation groups, the replicate 0 degree simulations had sufficiently different geometric sampling and unique lipid binding behavior that we though it should be used (4 us total) to achieve finer conformational resolution for each bnAb.

Figure S6 (now Figure 5-Figure Supplement 1)Please label the CDRs in C and provide a color key like in other figures. Also, please label the y-axes. This figure could move to main below 5B with the clusters "A,B,C" labeled on 5B.

We have added the axes labels and color key legend. We retained a minimal CDR loop labeling scheme for the more throughput interaction profiles here where colored sections in the residue axes denote CDR loop regions.

Figure S7 (Figure 7 Figure Supplement 1)Panels A and B would likely read better if swapped.

We have swapped these panels for a better flow.

For panel C, please display mean and standard deviation, and compare these values with an appropriate statistical test.

This is already displayed in main figure, we have removed it from supplement.

For E and F, please clarify from which trajectory(s) you are extracting this conformation from. Are these the global mean/representative poses? How do they compare to other geometrically distinct clusters?

The requested information was added to supplemental figure caption. These are frames from 2 distinct time points selected phosphate bound frames from 0-degree tilt replicates for both 4E10 and 10E8, representing at least 2 distinct macroscopic substates differing in global light chain and heavy chain orientation towards the membrane.

Table S2 (now Supplementary Table 3)Please add details for the 13h11 simulation.Additionally, please add average contact time and their standard deviation to the table, rather than just the aggregated total time. This will highlight the variability associated with the random initializations of each simulation.

We have added the details for 13h11 and the requested analysis (average aggregated time +/- standard deviation and average time per association event +- standard deviation) to supplement our summary statistics for this method.

Reviewer #2 (Recommendations For The Authors):(1) The structure of the manuscript should be improved. For example, almost half of the introduction (three paragraphs) summarize the results. I found it hard to navigate all the data and specific interactions described in the result section. Furthermore, the claims at the end of several sections seem unsupported. Especially for the generalization of the approach. This should be moved to the discussion section. The discussion is pretty general and does not provide much context to the results presented in this study.

We have significantly reorganized the results section to improve the flow of the manuscript and accessibility for readers, especially the first sections of all-atom simulations. We also removed claims not directly supported by data from our results, and expanded on some of these concepts in the discussion to make some more novel context to the result.

(2) The author should cite more rigorously previous work and refrain from using the term "develop" to describe the simple use of a well established method. E.g. Several studies have investigated membrane protein interactions e.g. [1], membrane protein-bilayer self-assembly [2], steered molecular dynamics [3], etc.

Thank you for identifying relevant work for the simulations that set precedent for our novel application to antibody-membrane interactions. We have removed language about development of simulation methods from the text and now better reference the precedent simulation methods used here.

(3) Have the authors considered estimating the PMF by combining the steered MD simulation through the application of Jarzynski's equality?

We performed from preliminary PMFs for Fab-membrane binding, but saw it was taking upward of 40 us to reach convergence. Steered simulations focus on a key lipid may be easier.

Although PMFs are beyond the scope of this work, we added proposals & allusion to their utility as the next steps for more rigorous quantification of fab-membrane interactions.

Minor(4) The term "integrative modeling" is usually used for computational pipelines which incorporate experimental data. Multiscale modeling would be more appropriate for this study.

We altered descriptions throughout the manuscript to reflect this comment.

(5) Units to report the force in the steered molecular dynamics are incorrect. They should be 98.

We changed axes and results to correctly report this unit.

(6) Labels for axes of several graphs are not missing.

We added labels to all axes of graphs, except for a few where stacked labels can be easily interpreted to save space and reduce complexity in figures.

(7) Figure 3 K & L is this really < 1% of total? The term "total" should also be clarified.

Thank you for pointing this out, we changed the % labels to be correct with axes from 0-100%. We clarified total in the figure caption.

(8) The font size in figures should be uniformized.

This suggestion has been applied

(9) Time needed for steered MD should be reported in CPUh and not hours (page 17).

We removed comments on explicit time measurements for our simulations.

(10) Version of Martini force field is missing in methods section

We used Martini 2.6 and added this to the methods.

References

(1) Prunotto, Alessio, et al. "Molecular bases of the membrane association mechanism potentiating antibiotic resistance by New Delhi metallo-β-lactamase 1." ACS infectious diseases 6.10 (2020): 2719-2731.

(2) Scott, Kathryn A., et al. "Coarse-grained MD simulations of membrane protein-bilayer self-assembly." Structure 16.4 (2008): 621-630.

(3) Izrailev, S., et al. "Computational molecular dynamics: challenges, methods, ideas. Chapter 1. Steered molecular dynamics." (1997).